# Neural dynamics of phoneme sequences reveal position-invariant code for content and order

Laura Gwilliams [1,2,3] ✉, Jean-Remi King [2,4], Alec Marantz[2,3,5,7] & David Poeppel[2,6,7]

Speech consists of a continuously-varying acoustic signal. Yet human listeners experience it as sequences of discrete speech sounds, which are used to recognise discrete words. To examine how the human brain appropriately sequences the speech signal, we recorded two-hour magnetoencephalograms from 21 participants listening to short narratives. Our analyses show that the brain continuously encodes the three most recently heard speech sounds in parallel, and maintains this information long past its dissipation from the sensory input. Each speech sound representation evolves over time, jointly encoding both its phonetic features and the amount of time elapsed since onset. As a result, this dynamic neural pattern encodes both the relative order and phonetic content of the speech sequence. These representations are active earlier when phonemes are more predictable, and are sustained longer when lexical identity is uncertain. Our results show how phonetic sequences in natural speech are represented at the level of populations of neurons, providing insight into what intermediary representations exist between the sensory input and sub-lexical units. The flexibility in the dynamics of these representations paves the way for further understanding of how such sequences may be used to interface with higher order structure such as lexical identity.

Speech comprehension involves mapping variable continuous acoustic signals onto discrete linguistic representations[1]. Although the human experience is typically one of effortless understanding, uncovering the computational infrastructure that underpins speech processing remains a major challenge for neuroscience[2] and artificial intelligence systems[3–5] alike.

Existing cognitive models primarily explain the recognition of words in isolation[6–8]. Predictions of these models have gained empirical support in terms of neural encoding of phonetic features[9–12], and interactions between phonetic and (sub)lexical units of

representation[13–16]. What is not well understood, however, is how sequences of acoustic-phonetic signals (e.g. the phonemes k-a-t) are assembled during comprehension of naturalistic continuous speech, in order to retrieve lexical items (e.g. cat).

Correctly parsing auditory input into phoneme sequences is computationally challenging for a number of reasons. First, there are no reliable cues for when each meaningful unit begins and ends (e.g. word or morpheme boundaries). Second, adjacent phonemes can acoustically blend into one another due to co-articulation[1]. Third, the same set of phonemes can form completely different words (e.g. pets

[1]Department of Neurological Surgery, University of California, San Francisco, USA. [2]Department of Psychology, New York University, New York, USA. [3]NYU Abu Dhabi Institute, Abu Dhabi, UAE. [4]École normale supérieure, PSL University, CNRS, Paris, France. [5]Department of Linguistics, New York University, New York, USA. [6]Ernst Strüngmann Institute for Neuroscience, Frankfurt, Germany. [7]These authors contributed equally: Alec Marantz, David Poeppel. ✉e-mail: leg5@nyu.edu

versus pest), which makes the order of phonemes—not just their identity—critical for word recognition.

The neurophysiology of speech-evoked activity also poses practical challenges. Previous studies have shown that speech sounds elicit a complex cascade of brain responses, spanning early auditory cortex, parietal and prefrontal areas[5,9,17]. Critically, the neural responses to an individual phoneme are long-lasting and greatly exceed the duration of the phoneme itself[11,16,18,19]. For instance, Gwilliams et al. (2018) found that properties of a speech sound are neurally encoded for more than 500 ms after the speech sound has dissipated from the acoustic signal. This phenomenon entails that a given phoneme is still encoded in neural activity while subsequent phonemes stimulate the cochlea. How does the brain reconcile auditory inputs that unfold more rapidly than the associated neural processes?

To identify how the brain represents continuous sequences of phonemes, we recorded 21 subjects with magneto-encephalography (MEG) while they attentively listened to 2 h of spoken narratives. We then use a combination of encoding and decoding analyses time-locked to each phoneme onset to track how the brain represents past and present phonetic features. In this work, we show how the language system (i) processes the phonetic features of multiple phonemes in parallel; (ii) keeps track of the relative order of those phonemes; and (iii) maintains information sufficiently long enough to interface with (sub)lexical representations. Our findings shed light on how both sequence order and phonetic content can be accurately represented in the brain, despite the overlapping neural processes elicited by continuous speech.

## Results

Neural responses were recorded with magnetoencephalography (MEG) while 21 participants listened to four short stories recorded using synthesised voices from Mac OS text-to-speech. Each participant completed two one-hour recording sessions, yielding brain responses to 50,518 phonemes, 13,798 words and 1,108 sentences per participant. We annotated each phoneme in the story for its timing, location in the word, as well as phonetic and statistical properties (Fig. 1A). The data have been uploaded to a public repository[20].

### Phonetic feature encoding in acoustic and neural signals

How are speech features represented in brain activity? To address this issue, we applied a two-step analysis. First, we dealt with the confound of speech features being correlated with low-level properties of the speech signal, such as pitch and intensity. We trained a temporal receptive field (TRF) model to predict MEG data from the amplitude envelope and pitch contour[21], and used this model to regress from the MEG signal all variance associated with these acoustic properties. All subsequent analyses were applied to the residuals of this model, and so can be interpreted with the potential confounds of loudness and pitch being removed.

Second, to track the neural representations of linguistic features, we fit a back-to-back regression (B2B)[22] to (1) decode 31 linguistic features from the residual MEG signal and (2) evaluate their actual contribution to the corresponding decoding performance. This was applied at each time sample independently. The suite of features included 14 acoustic-phonetic features (binary one-hot encoding of place, manner, and voicing), 3 information theoretic measures (surprisal, entropy, and log sequence frequency), 8 unit boundary features (onset/offset of words, syllables, morphemes) and 6 positional features (phoneme and syllable location in syllable, word, and sentence). All features were included in the model simultaneously. B2B was chosen to control for the co-variance between features in the multivariate analysis while optimising the linear combination of MEG channels to detect the encoding of information even in low signal-to-noise circumstances[22]. We applied the same model separately to the MEG signals ($n = 208$ channels) and to the acoustic signal ($n = 208$

frequency bands of the acoustic mel spectrogram, which approximates the auditory signal after passing through the cochlea[23]).

The output of the B2B is a set of beta coefficients that map a linear combination of ground truth features to the decoder's predictions of a feature. For explicit comparison between the results using more classic decoding metrics (AUC) and B2B regression coefficients, see Supplementary Fig. 1. For reference, the average significant decoding accuracy is very modest, at around 51–52%, where chance level is 50%. The effect sizes being dealt with here, therefore, are very small. But given the number of phonemes presented to the participants, our results are highly significant: all $p < 10^{-4}$. The effect sizes we report correspond to the proportion of variance explained by each feature according to the B2B model, and the relative noise ceiling estimate across all features. The noise ceiling is estimated as the maximum amount of variance the model was able to explain at any latency. We compute this by summing beta coefficients across all features and taking the maximum over time. Each feature timecourse is then normalised by this maximum performance measure, to provide a proportion of variance explained (see Methods).

An important theoretical point is that we are decoding features of each phoneme rather than decoding phoneme categories per se. However, when decoding phoneme categories instead of features, we observe very similar results (see supplementary Fig. 14).

Figure 1B summarises the result of the neural analysis. Confirming previous studies[9,11,16], phonetic features were decodable on average between 50–300 ms from the MEG signal and accounted for 46.2% of variance explained by the full suite of 31 features. Performance averaged across features was statistically greater than chance, as confirmed with a temporal permutation cluster test, based upon a one-sample t-test (df=20) applied at each time-sample ($p < 0.001$; critical t averaged over time = 3.61). Information theoretic measures such as surprisal, entropy, and log sequence frequency accounted for 14.5% of explainable variance, which was also statistically better than chance on average ($-10$–540 ms; $p < 0.001$; $t = 2.98$). Positional properties such as location in the word explained 31% of variance, and showed the greatest individual effect sizes ($-120$–600 ms; $p < 0.001$; $t = 5.61$). Here, phoneme order is coded the same for all phonemes regardless of feature. Finding a significant effect of phoneme order therefore suggests that order is represented independently from phonemic content, and is the same for all phonemes regardless of the features they contain. The remaining variance was accounted for by boundary onset/offset features (0–410 ms; $p < 0.001$; $t = 3.1$).

Are the phonetic feature representations contained in the neural signal similar to those contained in the auditory input? Fig. 1C shows the result of the same B2B analysis, but now applied to the audio mel spectrogram. Phonetic features accounted for 52.1% of the explainable variance between 0–280 ms ($p < 0.001$; $t = 9.56$). We also observed correlates of statistical information theoretic measures ($-180$–140 ms; $p < 0.01$; $t = 2.92$) and phoneme position ($-200$–580 ms; $p < 0.001$; $t = 4.59$). Phonetic features that were better decoded from the acoustic signal were also better decoded from the neural signal (Spearman correlation of average performance $r = 0.59$; $p = 0.032$). However, there was no significant correlation between the decoding performance of information theoretic measures or phoneme position across the auditory and neural analyses (Spearman $r = 0.13$; $p = 0.41$), and these features accounted for significantly more variance in the MEG as compared to the auditory analysis ($t = 2.82$; $p = 0.012$). Overall, this analysis suggests that phonetic and statistical features of speech have correlates with the acoustic signal, but that the decoding of higher-order structure from neural responses cannot be explained by the acoustic input alone.

### Multiple phonemes are processed simultaneously

Our finding that the average duration of phonetic decodability is around 300 ms is important, because this greatly exceeds the average

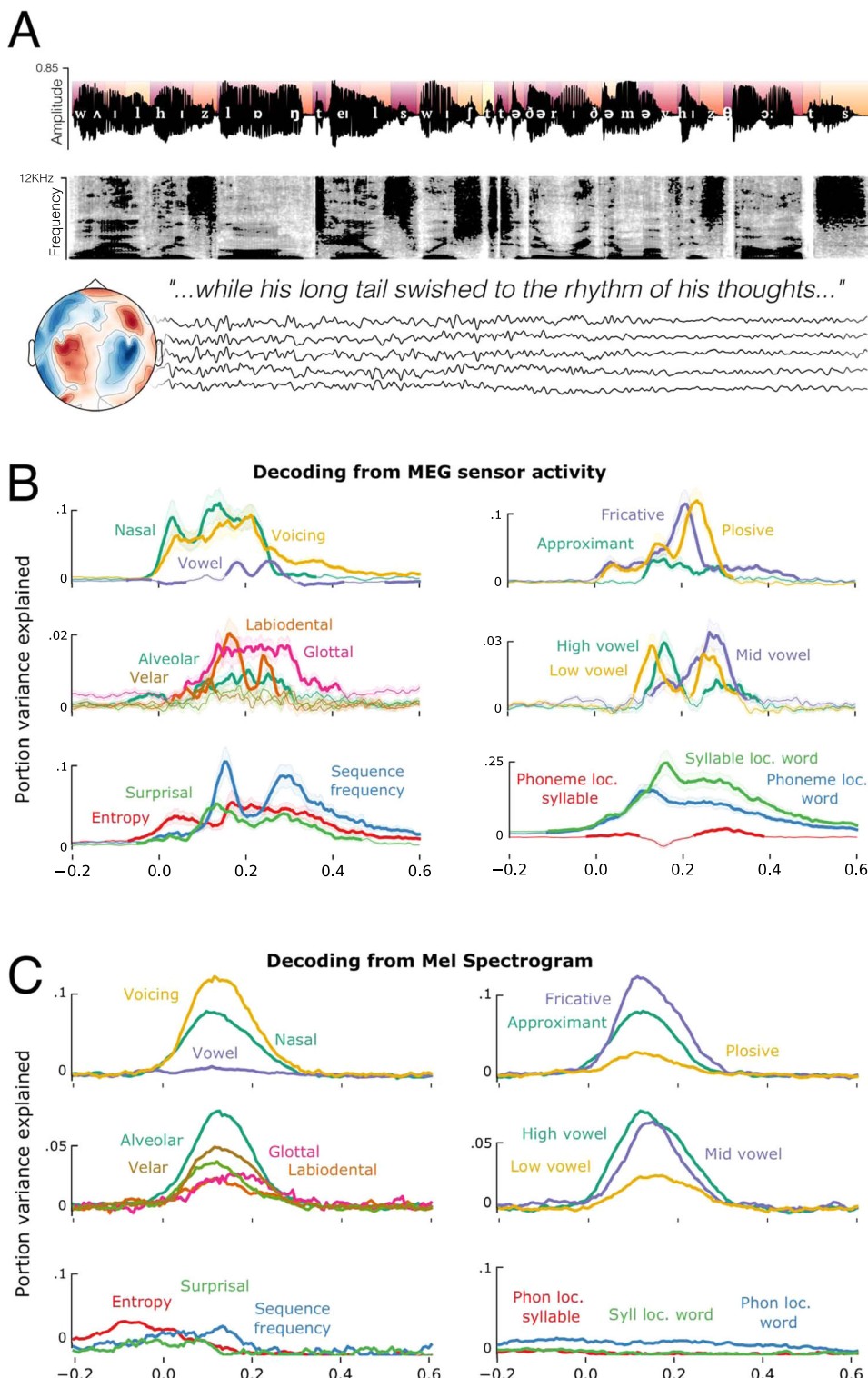

**Fig. 1 | Experimental design and phoneme-level decoding. A** Example sentence, with the phonological annotation superimposed on the acoustic waveform. Colours of the segments at the top half of the waveform indicate the phoneme's distance from the beginning of a word (darker red closer to word onset). The spectrogram of the same sentence appears below the waveform. Five example sensor residual time-courses are shown below, indicating that all recordings of the continuous stories were recorded with concurrent MEG. **B** Timecourses of decoding phoneme-level features from the MEG signal using back-to-back regression, time-locked to phoneme onset. Traces represent the average decoding performance over the 21 participants, shading represents standard error of the mean over participants. **C** Timecourses of decoding the same features from the mel spectrogram. The y-axis represents portion of explainable variance given all features in the model. Source data are provided as a Source Data file.

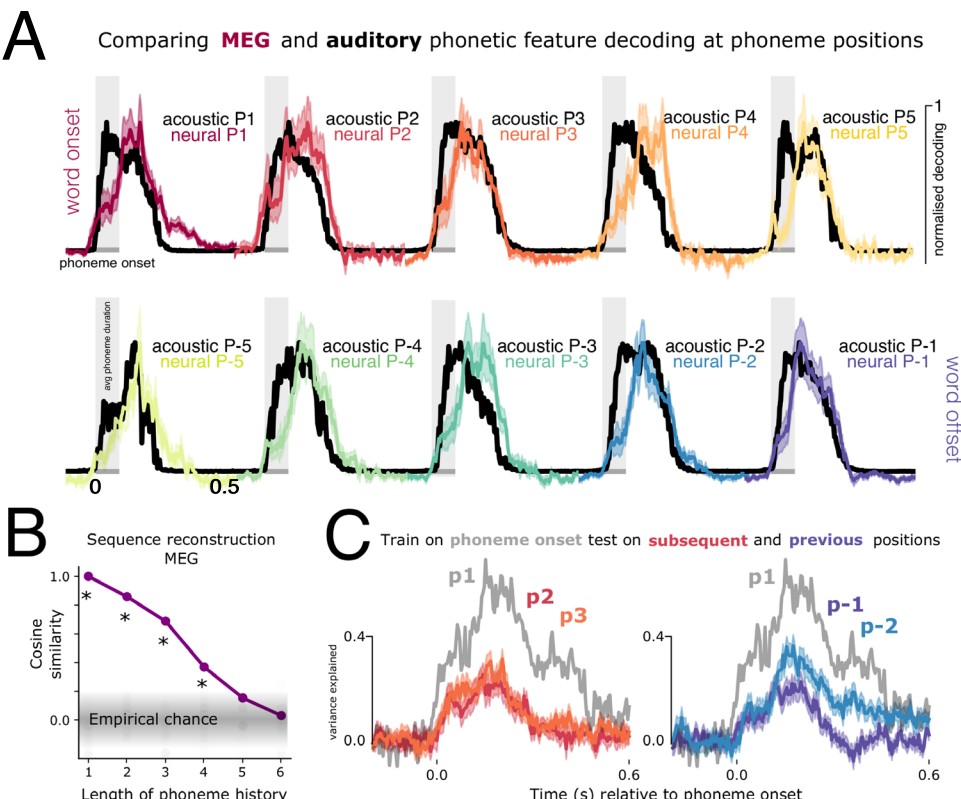

**Fig. 2 | Decoding as a function of phoneme position. A** Timecourses of decoding phoneme-level features from the mel spectrogram (black) and MEG sensors (coloured), time-locked to phoneme onset. Performance is averaged over the 14 phonetic feature dimensions. Shading in the neural data corresponds to the standard error of the mean across the 21 participants. Results are plotted separately for 10 different phoneme positions, where P1:P5 indicates distance from word onset (red colour scale) and P-1:P-5 distance from word offset (purple colour scale). E.g. P-1 is the last phoneme of the word, P-2 is the second to last phoneme of the word. *Y*-axis shows the beta coefficients normalised between 0–1. All plots share the same *y*-axis scale. **B** Result of simulating and reconstructing phoneme sequences from the MEG model coefficients. *x*-axis represents how many previous phonemes can be reconstructed; *y*-axis represents the cosine similarity between the true phoneme

sequence and the reconstruction. History lengths exceeding 4 phonemes correspond to the phonemes of the previous word in the sequence. All positions with an asterisk are significant at $p < 0.001$, derived from comparing the true value to a random permuted distribution. **C** Testing generalisation across phoneme positions. Grey trace shows average performance when training on phonemes at word onset and testing on phonemes at word onset. Coloured traces show average performance when training on phonemes at word onset, and testing at second (P2), third (P3), last (P-1) and second to last (P-2) phoneme positions. Shading represents the standard error of the mean across participants. *Y*-axis is the noise-normalised variance explained within the generalisation analysis. Source data are provided as a Source Data file.

duration of the phonemes in our stories (78 ms, SD = 34 ms). This implies that the brain processes multiple phonemes at the same time. To explicitly test how many phonemes are processed simultaneously, we decoded, from the same MEG responses, the place, manner, and voicing of the current and three preceding phonemes. We find that the phonetic features of a 3-phoneme history can be robustly decoded from an instantaneous neural response (see Supplementary Fig. 9).

To further assess whether this result implies that the brain can retrieve the content of phoneme sequences, we simulated the neural responses to continuous sequences of 4-phoneme anagrams using the decoding model coefficients to reconstruct synthesised MEG responses (see Methods). Under ideal conditions (no added noise) neural responses robustly encoded the history of four preceding phonemes (Fig. 2B). This result confirms the high degree of overlapping content during phoneme processing. Although decodability need not imply that the brain actually uses this information[24], this result indicates that the brain has access to at least a 3-phoneme sequence at any given processing time.

**Phonetic feature representations are position invariant**
Having demonstrated that the brain processes multiple speech sounds at the same time, the next question is: How does the brain do this without mixing up the phonetic features of these speech sounds? There are a number of potential computational solutions to this

problem. One is position-specific encoding, which posits that phonetic features are represented differently depending on where the phoneme occurs in a word. This coding scheme uses a different neural pattern to encode information about the first phoneme position (P1), second phoneme position (P2), etc., resulting in no representational overlap between neighbouring speech sounds.

To test whether the brain uses this coding scheme, we trained a decoder on the responses to phonemes in first position and evaluated the model's generalisation to other phoneme positions (Fig. 2C). Contra to the predictions of a position-dependent coding scheme, we found significant generalisation from one phoneme position to another. A classifier trained on P1 significantly generalised to the pattern of responses evoked by P2, P3, P-1 and P-2 from 20 to 270 ms ($p < 0.001$; $t = 3.3$), with comparable performance (max variance for P2 = 26%, SEM = 4%; P3 = 32%, SEM = 3%; P-1 = 23%, SEM = 3%, P-2 = 37%, SEM = 4%). This result contradicts a purely position-specific encoding scheme, and instead supports the existence of a position-invariant representation of phonetic features.

Interestingly, training and testing on the same phoneme position (P1) yielded the strongest decodability (max = 71%, SEM = 5%), which was significantly stronger than when generalising across positions (e.g. train P1 test P1 vs. train on P1 test on P2: 110:310 ms, $p = 0.006$). It is unclear whether this gain in performance is indicative of position-specific encoding in addition to invariant encoding, or whether it

reflects bolstered similarity between train and test signals due to matching other distributional features. Future studies could seek to match extraneous sources of variance across phoneme positions to test this explicitly.

## Phonetic representations rapidly evolve over time

The sustained, parallel and position-invariant representation of successive phonemes leaves our original question unanswered: How does the brain prevent interference between neighbouring speech sounds?

A second possible solution to this problem is that, rather than embedding phonetic features within a different neural pattern depending on phoneme location, phonetic features are embedded in different neural patterns as a function of latency since phoneme onset. This means that each speech sound travels along a processing trajectory, whereby the neural population that encodes a speech sound evolves as a function of elapsed processing time. To test whether this is the case, we used temporal generalisation (TG) analysis[25]. This method consists of learning the most informative spatial pattern for a phonetic feature at a given moment, and assessing the extent to which it remains stable over time. TG results in a training × testing matrix that helps reveal how a neural representation evolves over time. This analysis is applied relative to phoneme onset.

To evaluate the extent of representational overlap across phoneme positions, we aligned the TG matrices relative to the average latency between two adjacent phonemes. For example, relative to word onset, phoneme P1 is plotted at $t = 0$, P2 at $t = 80$, P3 at $t = 160$, and so on (Fig. 3). Note that the number of trials at test time decreases as we test phoneme positions further from word boundaries, due to differences in word length. P1 and P-1 are defined for 7335 trials per participant per repetition, P2 and P-2 for 7332 trials, P3 and P-3 for 5302 trials, P4 and P-4 for 3009 trials, P5 and P-5 for 1744 trials. Models are trained on all phonemes regardless of position (25259 per participant per repetition). This encourages the model to learn the topographic pattern which is common across phoneme positions.

While each phonetic feature could be decoded for around 300 ms (train time == test time (diagonal axis), Fig. 3A), the corresponding MEG patterns trained at each time sample were only informative for about 80 ms (marginal over train time (horizontal axis), Fig. 3A). This discrepancy demonstrates that the neural representations of phonetic features changes over time. We confirmed the statistical reliability of this evolution using an independent samples t-test, comparing diagonal and horizontal decoding performance axes (df = 200, $p < 0.001$; $t = 7.54$).

This analysis demonstrates that the neural representation of phonetic features indeed varies with latency since phoneme onset, suggesting that the brain uses a dynamic coding scheme to simultaneously represent successive phonemes.

## Dynamics encode the order of phonemes in sequences

The neural pattern that encodes phonetic features evolves with elapsed time since phoneme onset. What purpose does this evolution have? We sought to test whether the evolution is systematic enough to explicitly encode latency since the speech sound begun. If latency information is indeed encoded along with phonetic detail, the consequence would be an implicit representation of phoneme order in addition to phoneme identity. This is because the brain would have access to that fact that features of phoneme X occurred more recently than than phoneme X - 1, and those, more recently than phoneme X - 2, etc.

To formally test this possibility, we cropped all phoneme-aligned responses between 100 and 400 ms and used ridge regression to decode the latency since phoneme onset (see "Methods"). The reconstructed latency significantly correlated with the true latency (Pearson $r = 0.87$, $p < 0.001$; see Fig. 4A), suggesting that the dynamic spatial topography encodes the time elapsed since phoneme onset, in

addition to phonetic content. This thus distinguishes sequences composed of the same phonemes such as 'pets' versus 'pest', because the features of the speech sounds are time-stamped with the relative order with which they entered into the system. A crucial question to be addressed in future work is whether behavioural confusions in language perception arise from errors in these time-stamps.

## Dynamic coding separates successive phonetic representations

Next we asked whether another purpose of this dynamic coding scheme is to minimise overlap between neighbouring speech-sound representations. To test this, we quantified the extent to which successive phonemes are simultaneously represented in the same neural assemblies.

We extracted all time samples across train and test time where decoding performance exceeded a $p < 0.05$ threshold, Bonferroni-corrected across the 201 time samples of a single processing time. This is equivalent to a statistical $t$ threshold > 4, an effect size threshold variance explained > 5%, and decoding AUC > 0.507, as represented by the darker contour in Fig. 3. We then compared the number of significant time-points shared within and across phoneme positions. On average, 7.3% of time-points overlapped across phoneme positions (SD = 9%). The majority of overlap was caused by the first phoneme position, which shared 20% of its significant time-samples with the preceding phonemes, and reciprocally, the last phoneme which shared 27.9% of its time-samples with the subsequent (as can be seen in the left-sided appendage that overlaps with the last phoneme of the previous word). When removing the first and last phonemes from the analysis, this overlap drops to just 3.1% (SD = 3.2%).

Overall, these results show that while multiple phonemes are processed in parallel, any given pattern of neural activity only represents the features of one phoneme at a time, granting each phoneme an individuated representation. This avoids representational overlap between the phonetic features of neighbouring speech sounds. Future work would benefit from linking behavioural difficulty in speech comprehension with the extent of representational overlap, to assess the existence of a causal relationship.

## Dynamic representations are absent from audio spectrograms

Is this dynamic coding scheme also present in the auditory signal, and so all of this comes for free by virtue of speech being a dynamic input? Or, is dynamic encoding the result of a specific transformation the brain applies to that input? To address this issue, we applied the TG analysis to the audio mel spectrogram. We found no statistical difference between the accuracy time-course of a single decoder as compared to independent decoders at each time sample ($p = 0.51$; $t = -0.67$). The 'square' shape of the TG matrix suggests that acoustic signals contain stationary cues for acoustic-phonetic features (see Supplementary Fig. 7). We applied the overlap analysis to the auditory signal, using the same statistical threshold to identify relevant time-samples. On average, 92.5% of time-samples overlapped across positions (SD = 12.3%). This remained similar when removing the first and last phonemes from the analysis (mean = 91%, SD = 13.3%). The representational overlap in auditory encoding was significantly greater than neural encoding ($t = -21.3$, $p < 0.001$). We also confirmed that this overlap was qualitatively similar in different signal-to-noise ratio simulations (see Supplementary Fig. 2).

Together, these results demonstrate that the dynamic coding scheme we observe in neural responses is not a trivial reflection of the acoustic input. Rather, it is the consequence of an active process the brain applies to the relatively static auditory representations.

## MEG topographies associated with phonetic representations

To clarify the spatial underpinning of these dynamic representations, we analysed the beta coefficients of the decoding model when withholding regularisation (see "Methods"). One important question is

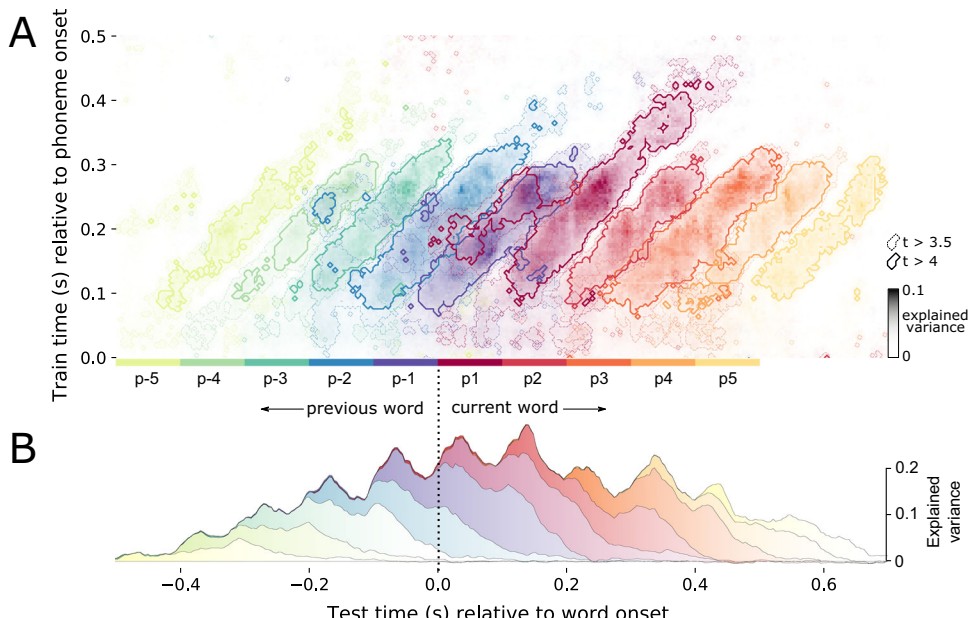

**Fig. 3 | Phonetic feature processing across the sequence. A** Temporal general-isation (TG) results superimposed for 10 phoneme positions. Results represent decoding performance averaged over the 14 phonetic features. From the first phoneme in a word (P1, dark red) moving forwards, and the last phoneme in a word (P-1, dark blue) moving backwards. The result for each phoneme position is shifted by the average duration from one phoneme to the next. The *y*-axis corresponds to the time that the decoder was trained, relative to phoneme onset. The *x*-axis cor-responds to the time that the decoder was tested, relative to word onset. Contours represent a *t*-value threshold of 4 (darker lines) and 3.5 (lighter lines). Note that fewer trials are entered into the analysis at later phoneme positions because words contain different numbers of phonemes. This is what leads to the reduction of decoding strength at p5 and p-5, for example. **B** Decoding performance of just the diagonal axis of each phoneme position (where train times are equal to test times). The visualisation therefore represents when phonetic information is available, regardless of the topography which encodes it. We use a stack plot, such that the variance explained by different phoneme positions is summed along the y-axis. Source data are provided as a Source Data file.

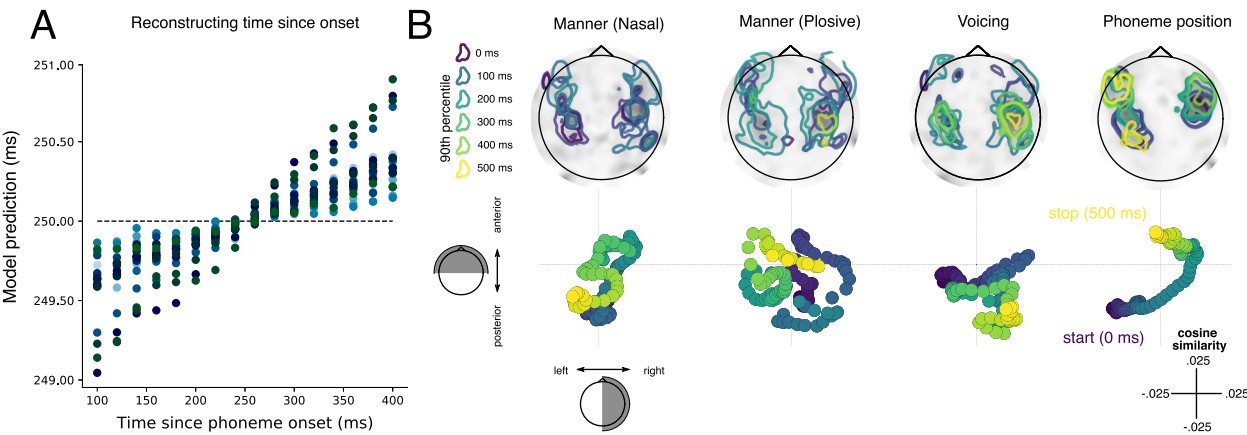

**Fig. 4 | Spatial evolution of processing trajectory. A** Reconstructing time since phoneme onset from 15 equally spaced samples from 100 to 400 ms. The hor-izontal dashed line indicates mean of time latency (i.e. what would be recon-structed in absence of evidence to the contrary). The coloured dots represent the 21 participants at a given time sample. **B** Decoding model coefficients plotted as topographies (above) and trajectories (below) over time. Coefficients are shown for the decoding of three phonetic features and phoneme position. Phoneme onset is shown in the darkest purple (0 ms) and colours shift towards yellow as a function of time. Contours on the topographies highlight 90% percentile coefficient magnitude at 100 ms increments. Trajectories below represent the cosine distance between the coefficient topography and a binary posterior/anterior mask (*y*-axis) and a binary left/right mask (*x*-axis). Overall the results show that the topographies evolve over time, where phonetic features remain concentrated around auditory cortex, and phoneme position takes an anterior path from auditory to frontal cortices. Source data are provided as a Source Data file.

whether phonetic representations remain spatially local, and evolve within auditory regions, or whether they shift globally over space; for example, from auditory to frontal cortices.

Figure 4B displays the temporal evolution of the MEG topo-graphies associated with the most robustly encoded phonetic features. We quantified the extent of spatial evolution over time by projecting the coefficients onto orthogonal axes of the sensor array (front/back, left/right) and computing the extent of temporal structure (see Methods for details).

Overall, we find that activity which encodes phonetic detail remains local within auditory regions, in contrast to the representation of phoneme position which followed a clear posterior-anterior

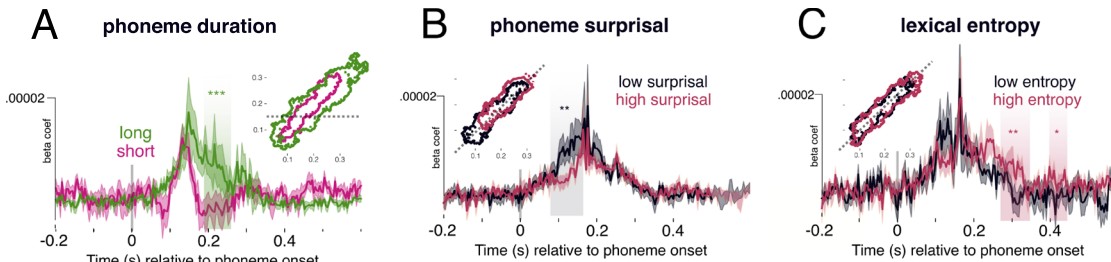

**Fig. 5 | Manipulating axes of sequence dynamics.** We find that different properties of the speech input modulate decoding latency, width and angle. **A** TG analysis median split into phoneme length: short (average 45 ms) and long (average 135 ms). Contour inlay represents the borders of the significant temporal clusters at p < 0.001, derived from a non-parametric randomised permutation procedure. Waveforms represent a horizontal slice at 140 ms (shown as a dashed line in the contour plot). **B** Analysis on all non-onset phonemes split into median surprisal, along the diagonal plane (slice shown in contour plot). Highlighted areas show significant temporal clusters that distinguish low and high surprisal, derived from a permutation cluster test (exact p-value 0.007). **C** Analysis on all non-onset phonemes split into median cohort entropy, also along the diagonal plane. Highlighted areas show significant temporal clusters between low and high entropy, derived from a permutation cluster test (exact p-values .002 and 0.02). Shading on the waveform of all plots represents standard error of the mean across the 21 participants. *p < 0.05; **p < .01; ***p < 0.001, when comparing true values to 1000 random permuted test statistics. Source data are provided as a Source Data file.

gradient. Both of these neural patterns encoding phonetic features exhibited significantly more structured movement over time than a null trajectory (p < 0.001). Given that the brain has simultaneous access to phoneme content and elapsed processing time, the coupling of these trajectories permits simultaneous read-out of both the identity and the relative order of a speech sound in a sequence from any given spatial pattern of activity.

### Representations evolve at the rate of phoneme duration

Although phonetic features are present in neural responses for around 300 ms (length of the decoding diagonal in Fig. 3A), any given spatial pattern of activity which encodes that information (width of the decoding diagonal) is informative for much shorter than that – about 80 ms. We noted that 80 ms is very similar to the average duration of the phonemes in our stories (M = 78 ms, SD = 34 ms). A crucial question is whether the speed of evolution is fixed, or whether it scales with phoneme duration and speech rate.

We grouped trials into quartiles and analysed brain responses to the shortest and longest phonemes (~4500 trials in each bin; mean duration 45 and 135 ms, respectively). We tested whether there was a significant difference in the duration of generalisability between these two groups of trials, whereby shorter generalisation indicates faster spatial evolution, and longer generalisation slower spatial evolution. We found that longer phonemes generalised for an average 56 ms longer than shorter phonemes (p = 0.005; t = −2.6) (Fig. 5A), suggesting that phoneme duration is indeed an important factor for how quickly a spatial pattern evolves during processing.

In addition, we observed a small but robust deflection in the angle of the decoding diagonal. If a processing trajectory moves faster than the training data, it will have an angle of less than 45 degrees. Slower trajectories will have an angle greater than 45 degrees. Using principal component analysis to find the axis of greatest variance, we found that shortest phonemes had an average angle of 42.3 degrees, whereas the longest phonemes had an average angle of 47.1. This difference was statistically significant when applying an independent samples t-test between short and long phonemes across participants (df = 20, t = 2.56, p = 0.013).

This pattern suggests that the brain adapts its speed of processing as a function of the rate of speech input, to ensure that approximately the same number of phonemes are encoded at a given test time while minimising representational overlap. This potentially supports the conjecture that the brain preferentially derives higher order structure by a computing sliding triphone sequences from the speech input.

### Phonetic representations vary with linguistic processing

To what extent do the neural representations of phonetic features inform linguistic processing? To address this question, we next tested whether the dynamics of phonetic representations systematically vary with (i) word boundaries, (ii) surprisal, and (iii) lexical identification.

We evaluated decoding performance at word boundaries: word onset (position P1) and word offset (position P-1) separately for each family of phonetic features (place of articulation, manner, and voicing). Phonetic features were decodable later at word onset than offset, yielding a significant difference during the first 250 ms (p's <0.03 for all features). When averaging decoding performance across phonetic features, the lag between neural and acoustic maximum accuracy was 136 ms (SD = 13 ms) at word onset, which reduced to 4 ms (SD = 13 ms) at word offset (see Fig. 2A). This average onset/offset latency difference was statistically significant (t = −3.08; p = 0.002). Furthermore, place and voicing features were sustained in the neural signal significantly longer for phonemes at the beginning of words as compared to the end, which was also true when averaging over all features (p < 0.001, t = −3.79, 328–396 ms) (see Fig. 3).

Is this latency shift purely lexical in nature, or could statistical properties of speech sounds at word boundaries explain the result? In general, and in our speech stimuli, phonemes at the beginning of words are less predictable than phonemes at the end of words. We tested whether expected phonemes could be decoded earlier from the neural signal than unexpected ones by grouping trials as a function of surprisal. To control for co-linearity between word-boundaries and surprisal, we selected all phonemes that were not at word onset, and tested decoding accuracy relative to quartile phoneme surprisal (see Methods for details on how this variable was computed). Each analysis bin contained ~4500 trials. If the shift in latency is purely a word-boundary effect, we should observe no latency difference as a function of non-onset phoneme surprisal.

We observed a systematic latency shift as a function of surprisal (Fig. 5B): more predictable phonemes were decoded earlier than less predictable ones, leading to a significant difference between low and high surprisal from 120–132 ms (p = 0.007). Surprisal did not significantly modulate peak decoding accuracy (all uncorrected p-values > 0.2).

This result suggests that the brain initiates phonetic processing earlier when the phoneme identity is more certain. This result at least partially, and potentially fully, explains the latency shift at word boundaries.

Our results suggest that the temporal dynamics of phonetic processing are modulated by certainty about the phoneme unit being said. Does this extend to certainty about word identity, such that phonetic

representations are maintained until higher order structure can be formed?

We quantified word uncertainty using lexical cohort entropy[26]. If many words are compatible with a given phoneme sequence with a similar likelihood, then lexical entropy is high. On the contrary, if only one word is likely given a specific phoneme sequence (which is more often the case towards word offset), then lexical entropy is low. We evaluated whether decoding performance between 300–420 ms (the window that showed the word onset/offset effect) varied with lexical entropy (Fig. 5C). We grouped trials based on lexical entropy, and ran the analysis on all phonemes that did not occur at word onset (~4500 trials per bin). In the window of interest, higher entropy phonemes were decoded with significantly higher performance (304–328 ms, $p = 0.002$, $t = -2.12$). This suggests that phonetic information is maintained for longer in cases of higher lexical uncertainty.

When we expand the analysis window to the entire 0–500 ms epoch with a lower cluster-forming threshold, we find that lexical entropy significantly modulates phonetic decoding from 200–420 ms $p = 0.016$, $t = -2.61$. This is a reasonable time-frame within which lexical entropy could exhibit an effect, and matches the time-frame of entropy reduction in our stimulus materials remarkably well. The average entropy of the low-entropy bin was 1.19 bits, which on average occurs 333 ms into a word (SD = 158 ms). The average entropy of the high-entropy bin was 4.3 bits, which on average occurs 108 ms into a word (SD = 51 ms). This therefore means that the average time between a phoneme in the high entropy bin and low entropy bin was 225 ms, which matches the duration of the entropy modulation effect in our data (220 ms). This in turn suggests that phonetic detail is maintained until lexical ambiguity is sufficiently reduced.

In a separate analysis, we tested whether word length played a role in the maintenance of phonetic detail of the onset phoneme. We compared decoding performance of word-onset phonemes grouped into median word length (shorter mean length = 2.54 phonemes; 4058 trials; longer mean length = 5.2 phonemes; 2841 trials). No significant differences between groups were found (all clusters $p > 0.2$, duration <2 ms).

Overall, our results confirm that the neural representations of phonemes systematically vary with lexical properties.

## Discussion

How the brain processes sequences is a major neuroscientific question, fundamental to most domains of cognition[27,28]. Precise sequencing is particularly vital and central in speech comprehension: Sensory inputs are transient and unfold rapidly, yet the meaning they convey must be constructed over long timescales. While much is known about how the features of individual sounds are processed[9,11], the neural representation of speech sequences remains largely unexplored.

Here we analyse neural responses of human participants listening to natural stories, and uncover three fundamental components of phonetic sequencing. First, we add specificity to the claim that the brain does not process and discard inputs at the same rate with which new inputs are received[16]. Namely, we find that the phonetic representations of the three most recently heard phonemes are maintained in parallel. Second, we show how the brain reconciles inputs that unfold faster than associated neural processing: The content of a speech sound is jointly encoded with the amount of time elapsed since the speech sound began. This encoding scheme is what allows the brain to represent the features of multiple phonemes at the same time, while housing them within distinct activity patterns. Third, we observe that the timing of initiation and termination of phonetic processing is not fixed. Rather, processing begins earlier for more predictable phonemes in the sequence, and continues longer when lexical identity is uncertain. This has the critical implication that phonetic processing, phoneme sequence processing and lexical processing are engaged in continuous interaction. Our results provide insight into the temporal

dynamics of auditory sequence representations and associated neural computations, and pave the way for understanding how such sequences in speech index higher order information such as lexical identity.

Our analyses show that speech sound properties are neurally represented for much longer than the sensory input, permitting the auditory system access to the history of multiple (at least three—our lower bound estimate) phonemes simultaneously. Crucially, the activity pattern encoding these features evolves, systematically, as a function of elapsed processing time, which prevents consecutive speech sounds from co-occupying the same activity pattern. This neural coding scheme grants two computational advantages. First, it serves to avoid representational overlap between neighbouring speech sounds, thus preserving the fidelity of the content of phonetic representations. Second, the systematic spatial evolution of the neural pattern encodes amount of time since speech sound onset, and therefore the relative location of that sound in the sequence. The encoding of phonetic content regardless of order, and order regardless of content, allows the brain to represent running sequences of speech sounds. This phonetic trigram is a candidate intermediate representation between phonetic features and stored (sub)lexical representations.

Joint content-temporal coding resonates with recent evidence for dedicated temporal codes in rat hippocampus[29] and in the human visual system[30]. The processing trajectory we find for human speech processing did not trace a wide spatial path across distinct regions. Instead, phonetic features remained locally encoded within auditory regions for around 100–400 ms, and the trajectory was different for different features. This suggests that information may be locally changing its population-level configuration within the auditory cortices rather than following a strict anatomical transposition from low to high level areas[5,12,30]. Unfortunately, the signal strength of our single trial estimates, and the spatial resolution of MEG, limits the specificity of the spatial claims we can make based on these data. Future work will thus be critical to clarify how the location and configuration of these responses changes as a function of processing time.

Finding that phonetic content is encoded similarly at different phoneme positions rules out a number of competing hypotheses for how sequences are neurally represented. First, it is difficult to reconcile these results with an explicit sequence representation. For example, if the brain represents the sequence of all elapsed phonemes as a whole, the representation of phoneme X at word onset would generalise poorly to third position ABX and even worse to sixth position ABCDEX. Second, under the same logic, this result rules out the idea that phonemes have a purely context-dependent encoding scheme, such as being represented along with their co-articulatory neighbours as with 'Wickelphones'[31]. In that case, phoneme X would have a different representation in the context AXB and VXY. Finally, generalisability is inconsistent with position-specific encoding accounts, such as edge-based schemes[32,33], which would posit that X is encoded differently in ABX and XBC. While it is possible that multiple representational systems co-exist, our results support that at least one of those encoding schemes is context-independent, which encodes content regardless of lexical edges or phoneme neighbours.

Several aspects of our results suggest that the neural representations of phonetic features directly interact with high-level linguistic processing. First, the encoding of phonetic features is systematically delayed as a function of phonological uncertainty (surprisal) and systematically sustained as a function of lexical uncertainty (cohort entropy). These latency shifts fit with models of predictive coding[34,35] and analysis-by-synthesis[36]: when predictability for a phoneme is high, processes can be initiated earlier (perhaps in some cases before the sensory input) than when the phoneme identity is unknown. While previous work has shown that processing of the

speech signal is sensitive to phoneme probability within a word[13,15,37–39] (see[26] for a review), here we quantify the consequences this has for encoding the phonetic content of those phonemes. Processing delays may serve as a compensatory mechanism to allow more information to be accumulated in order to reach the overall same strength of encoding[40,41]. Future work should test whether this local (within-word) predictability metric has similar consequences to global (across-word) metrics.

Second, phonetic features of the sequence are maintained longer during lexical ambiguity. While previous work has shown active maintenance of phonetic information[16], here we provide evidence that maintenance is dynamic and scales with certainty over higher-order units. This result not only highlights the flexibility of speech processing but also demonstrates the bi-directional interaction between hierarchical levels of processing. Our results suggest that acoustic-phonetic information is maintained until (sub)lexical identity meets a statistically-defined threshold, providing clear processing advantages in the face of phonological ambiguity and lexical revision[16]. Future work would benefit from evaluating whether phonetic maintenance is implemented as a general working memory mechanism[42,43], in order to relate speech processing to the human cognition more broadly. Furthermore, our understanding would benefit from testing whether the target representations are lexical or morphological in nature, as well as orthogonalising lexical entropy and word-internal disambiguation points (i.e. when entropy reduces to zero) to more precisely quantify the temporal relationship with information maintenance.

Our study has several shortcomings that should be addressed in future work. First, the poor signal-to-noise ratio of single-trial MEG lead to very low decoding performances in our results—peaking only 1–2% greater than chance in most cases, on average. The results we report are detectable because we collected responses to thousands of speech sounds. This limits our ability to make claims about the processing of specific speech features, within specific contexts, which do not occur often enough in our dataset to sufficiently evaluate. Future work would benefit from recording a greater number of repetitions of fewer speech sounds to allow for signal aggregation across identical trials. Second, the spatial resolution and the signal-to-noise ratio of MEG remain insufficient to clearly delineate the spatial underpinning of the dynamic encoding scheme. Understanding how the location and configuration of neural responses to speech sounds evolves as a function of elapsed processing time will require data with both high spatial and temporal resolution, such as electro-corticography recordings from the cortical surface. Finally, in part to address the first limitation, we chose a passive listening paradigm, in order to optimise the number of trials we could obtain within a single recording session. This prohibits us from associating decoding performance with behaviour and task performance, which is a key next step for understanding the link between speech representations and speech understanding. A number of interesting hypotheses can be tested in this regard. For instance, are lapses in comprehension explained by poorer decoding of phonetic input? If the brain fails to transform information fast enough along the processing trajectory, does this lead to predictable errors in perception due to overlapping representations? Causally relating the representational trajectories we observe to successful and unsuccessful comprehension will be critical for further understanding the role of these computations for speech processing.

Our results inform what computational solution the brain implements to process rapid, overlapping phoneme sequences. We find that the phonetic content of the unfolding speech signal is jointly encoded with elapsed processing time, thus representing both content and order without relying on a position-specific coding scheme. The result is a sliding phonetic representation of the most recently heard speech sounds. The temporal dynamics of these computations are flexible, and vary as a function of certainty about both phonological and lexical identity. Overall, these findings provide a critical piece of the puzzle for how the human brain parses and represents continuous speech input, and links this input to stored lexical identities.

## Methods

### Ethical regulations
Our research was approved by the IRB ethics committee at New York University Abu Dhabi.

### Participants
Twenty-one native English participants were recruited from the NYU Abu Dhabi community (13 female; age: $M = 24.8$, SD = 6.4). All provided their informed consent and were compensated for their time at a rate of 55 Dirhams per hour. Participants reported having normal hearing and no history of neurological disorders. Each participant participated in the experiment twice. Time between sessions ranged from 1 day to 2 months.

### Stimulus development
Four fictional stories were selected from the Open American National Corpus[44]: Cable spool boy (about two bothers playing in the woods); LW1 (sci-fi story about an alien spaceship trying to find its way home); Black willow (about an author struggling with writer's block); Easy money (about two old friends using magic to make money).

Stimuli were annotated for phoneme boundaries and labels using the 'gentle aligner' from the Python module lower quality. Prior testing provided better results for lower quality than the Penn Forced Aligner[45].

Each of the stories were synthesised using the Mac OSX text-to-speech application. Three synthetic voices were used (Ava, Samantha, Allison). Voices changed every 5–20 sentences. The speech rate of the voices ranged from 145–205 words per minute, which also changed every 5–20 sentences. The silence between sentences randomly varied between 0–1000 ms.

### Procedure
Stimuli were presented binaurally to participants though tube earphones (Aero Technologies), at a mean level of 70 dB SPL. We used Presentation software to present the stimulus to participants (Neuro Behavioural Systems). The stories ranged from 8–25 min, with a total running time of ~1 h. Before the experiment proper, every participant was exposed to 20 s of each speaker explaining the structure of the experiment. This was designed to help the participants attune to the synthetic voices.

The order of stories was fully crossed using a Latin-square design. Participants heard the stories in the same order during both the first and second sessions.

Participants answered a two-choice question on the story content every ~3 min. For example, one of the questions was 'what was the location of the bank that they robbed'? The purpose of the questions was to keep participants attentive as well as to have a formal measure of engagement. Participants responded with a button press. All participants performed this task at ceiling, with an accuracy of 98%.

### MEG acquisition
Marker coils were placed at five positions to localise each participant's skull relative to the sensors. These marker measurements were recorded just before and after the experiment in order to track the degree of movement during the recording.

MEG data were recorded continuously, using a 208 channel axial gradiometer system (Kanazawa Institute of Technology, Kanazawa, Japan), with a sampling rate of 1000 Hz and an online low-pass filter of 200 Hz and a high-pass filter of 0.01 Hz.

## Preprocessing MEG

The raw MEG data were noise reduced using the Continuously Adjusted Least Squares Method (CALM: (Adachi et al., 2001)), with MEG160 software (Yokohawa Electric Corporation and Eagle Technology Corporation, Tokyo, Japan).

We used a temporal receptive field (TRF) model to regress from the raw MEG data responses that were sensitive to fluctuations in the pitch and envelope of the acoustic speech signal. We used the ReceptiveField function from MNE-Python[46], using ridge regression as the estimator. We tested ten lambda regularisation parameters, log-spaced between 1−6 and 1 + 6, and picked the model with the highest predictive performance averaged across sensors. MEG sensor activity at each ms was modelled using the preceding 200 ms of envelope and pitch estimates. Both the acoustic and MEG signals were processed to have zero mean across frequency bands and sensors respectively, and scaled to have unit variance before fitting the model. MEG acoustic-based predictions were then transformed back into original MEG units before regressing out of the true MEG signals. This process, including fitting hyper-parameters, was applied for each story recording and for each participant separately, across 3 folds. Because all analyses are fit on these residual data, we can interpret our results knowing that they cannot be accounted for by low-level acoustic attributes such as the acoustic amplitude.

The data were bandpass-filtered between 0.1 and 50 Hz using MNE-Python's default parame- ters with firwin design[46] and down-sampled from 1000 Hz to 250 Hz. Epochs were segmented from 200 ms pre-phoneme onset to 600 ms post-phoneme onset. No baseline correction was applied. Our results are not dependent upon the filter applied.

## Preprocessing auditory signals

We computed a time-frequency decomposition of the auditory signals by deriving a mel spectrogram representation using the Python module librosa (version 0.8.0). We applied a 2048 sample Hamming window to the auditory waveform, with a 128 sample overlap between successive frames. We derived a power estimate at each of 208 frequency bands (analogous to the 208 MEG channels) using non-linearly spaced triangular filters from 1−11250 Hz. These data were then also downsampled to 250 Hz, and segmented from 200−600 ms in order to match the dimensionality and size of the MEG epochs. No baseline correction was applied.

We also tried using 50 linearly spaced frequency bands, and this did not change the interpretation of our acoustic analyses.

## Modelled features

We investigated whether single-trial sensor-level responses varied as a function of fourteen binary phonetic features, as derived from the multi-value feature system reported in[47]. Note that this feature system is sparse relative to the full set of distinctive features that can be identified in English; however, it serves as a reasonable approximation of the phonemic inventory for our purposes.

Voicing. This refers to whether the vocal chords vibrate during production. For example, this is the difference between b versus p and z versus s.

Manner of articulation. Manner refers to the way by which air is allowed to pass through the articulators during production. Here we tested five manner features: fricative, nasal, plosive, approximant, and vowel.

Place of articulation. Place refers to where the articulators (teeth, tongue, lips) are positioned during production. For vowels, this consists of: central vowel, low vowel, mid vowel, high vowel. For consonants, this consists of: coronal, glottal, labial and velar.

Nuisance variables. In the same model, we also accounted for variance explained by 'nuisance variables' – i.e. structural and statistical co-variates of the phonemes. Though we were not interested in

interpreting the results of these features, we included them in the model to be sure that they did not account for our main analysis on the phonetic features. These features included: primary stress, secondary stress, frequency of the phoneme sequence heard so far, suffix onset, prefix onset, root onset, syllable location in the word, and syllable onset. These features were extracted from the English Lexicon Project[48].

Subset variables. Throughout the analysis, we subset trials based on their relationship to: word onset, word offset, surprisal, entropy, distance from onset, distance from offset.

Surprisal is given as:

$$P(p|C) = \frac{f(p)}{\sum_{p \in C} f(p)} \tag{1}$$

and cohort entropy is given as

$$-\sum_{w \in C} P(w|C)\log_2 P(w|C) \tag{2}$$

where **C** is the set of all words consistent with the heard sequence of phonemes thus far, and $f(w)$ is the frequency of the word $w$ and $f$(p) is the frequency of the phoneme $p$. Measures of spoken word frequency were extracted from the English Lexicon Project[48].

## Decoding

Decoding analyses were performed separately on the acoustic signal and on the neural signal. For the acoustic decoding, the input features were the power estimates at each of the 208 frequency bands from 1−11,250 Hz. For the neural decoding, the input features were the magnitude of activity at each of the 208 MEG sensors. This approach allows us to decode from multiple, potentially overlapping, neural representations, without relying on gross modulations in activation strength[49].

Because some of the features in our analysis are correlated with one another, we need to jointly evaluate the accuracy of each decoding model relative to its performance in predicting all modelled features, not just the target feature of interest. This is because, if evaluating each feature independently, we will not be able to dissociate the decoding of feature f from the decoding of the correlated feature $\hat{f}$.

Note that encoding models do not suffer from this issue, as they automatically disentangle the specific contribution of co-varying factors[50]. However, encoding models can be challenged by the low signal-to-noise of MEG, where each channels only capture a minute amount of signal, and where signals are best detected from a linear combination of MEG sensors.

To overcome the issue of feature co-variance, while still capitalising on the advantages of decoding approaches, we implemented a back-to-back (B2B) ridge regression model[51]. This model involves a two stage process.

In short, B2B consists of both a decoding and an encoding step: (1) the decoding model **G** f aims to find the combination of channels that maximally decode feature $f$ and (2) the encoding model $H_f$ estimates whether the decoded predictions $\hat{f}$ are specific to $f$ and/or attributable to other, covarying, features.

To implement B2B, we first fit a ridge regression model on a random (shuffled) 50% split of the epoched data. A decoding model $\mathcal{R}$ $G_f \in \mathcal{R}^C$ was trained across the C = 208 MEG channels for each of the F = 31 phonetic features independently at each of 201 time-points in the epoch. That is, we train and test a unique decoder at each time sample. The mapping was learnt between the multivariate input (activity across sensors) and the univariate stimulus feature (one of the 31 features described above). All decoders were fit on data normalised

by the mean and standard deviation in the training set:

$$argmin_G \sum_i (y_i - G^T X_i)^2 + \alpha_G |G|^2 \qquad (3)$$

where $y_i \in \{\pm 1\}$ is the feature to be decoded at trial $i$ and $X_i$ is the corresponding MEG activity. The l2-regularisation parameter $\alpha_G$ was also fit, testing 20 log-spaced values from 1−5 to 15. This was implemented using the RidgeCV function in scikit-learn[52].

Then, we use the remaining 50% split of data to train the encoder $\mathbf{H} \in \mathcal{R}^F$ for each of the decoded features. To do this, we fit a second ridge regression model to estimate the coefficients that map the true features onto the decoded features:

$$argmin_H \sum_i (Y_i - H^T G^T X_i)^2 + \alpha_H |H|^2 \qquad (4)$$

where $Y_i \in \{\pm 1\}$ are the true features of a given phoneme at trial i and $G^T X_i = \hat{Y}_i$ is the features decoded from the MEG at that trial. Again, a regularisation parameter $\alpha_H$ was learnt for this stage with RidgeCV.

This second step of B2B allows the models to estimate whether the decoded features f is solely explained by its covariates. In particular[51], showed that, under Gaussian and heteroscedasticity, the $H_f$, tends a positive value if and only if $f$ is linearly and specifically encoded in the brain activity and not reducible to its covarying features.

From this, we use the "beta" coefficient that maps the true stimulus feature to the predicted stimulus feature (diag($\mathbf{H}$)) as a metric of specific decoding performance. If a stimulus feature is not encoded in neural responses (the null hypothesis) then there will be no reliable mapping between the true feature $y$ and the model prediction $\hat{y}$. Thus, the beta coefficient will be indistinguishable from zero – equivalent to chance performance. If, however, a feature is encoded in neural activity (the alternative hypothesis), we should uncover a significant relationship between y and $\hat{y}$, thus yielding an above-zero beta coefficient.

This procedure was applied at all possible train/test time combinations for each time sample from −200 to 600 ms relative to phoneme onset. For the results shown in Figs. 1, 2 and 4, we just analyse the beta coefficients where the train time is equal to test time (the 'diagonal'). Whereas the analyses shown in Figs. 3 and 5 use the full 2-dimensional timecourse of decoding performance.

The train/test split was performed 100 times, and the beta-coefficients were averaged across iterations. This circumvents the issue of unstable coefficients when modelling correlated variables. These steps were applied to each participant independently.

## Proportion of variance explained

To aid interpretability of our effect sizes, we computed the proportion of variance explained for each feature. To obtain this measure, we analyse the beta coefficients that map each true feature $f$ to the target feature at each time sample $t$. Specifically, we first compute a noise ceiling, by summing the beta coefficients across all features, and taking the maximum value at any latency:

$$\hat{R}^f_{ceiling} = \max_t \sum_{f=1}^{f=31} \beta_f \qquad (5)$$

$\hat{R}_{ceiling}$ thus represents the upper limit of variance our model can explain, taking all features together. We then normalise the beta coefficient time-courses of each feature by this upper limit value:

$$\hat{R} = \hat{R}_{B2B} / \hat{R}_{ceiling} \qquad (6)$$

The result is a proportion of variance explained, relative to the contribution of all features in the model.

## Temporal generalisation decoding

Temporal generalisation (TG) consists of testing whether a temporal decoder fit on a training set at time $t$ can decode a testing set at time $t'$[25]. This means that rather than evaluating decoding accuracy just at the time sample that the model was trained on, we evaluate its accuracy across all possible train/testing time combinations. TG can be summarised with a square training time × testing time decoding matrix. To quantify the stability of neural representations, we measured the duration of above-chance generalisation of each temporal decoder. To quantify the dynamics of neural representations, we compared the mean duration of above-chance generalisation across temporal decoders to the duration of above-chance temporal decoding (i.e. the diagonal of the matrix versus its rows). These two metrics were assessed within each participant and tested with second-level statistics across participants.

## Comparing decoding performance between trial subsets

To compare decoding performance for different subsets of trials that are of theoretical interest (e.g. between high/low surprisal, or beginning/end of word), we add a modification to our train/test cross-validation loop. The data are trained on the entire training set (i.e. the same number of trials as the 'typical analysis'), and test set is grouped into the different levels of interest. We evaluate model performance separately on each split of the test data, which yields a time-course or generalisation matrix for each group of trials that we evaluate on.

## Group statistics

In order to evaluate whether decoding performance is better than chance, we perform second-order permutation-based statistics. This involves testing whether the distribution of beta coefficients across participants significantly differs from chance (zero) across time using a one-sample permutation cluster test with default parameters specified in the MNE-Python package[46]. The permutation test first computes the observed metric of interest, namely the summed t-value in a temporal cluster. The metric is then re-computed 10,000 times, each time randomly permuting the sign of the beta coefficients before finding temporal clusters. Significance is assessed by comparing the proportion of times that the summed t-value in any identified temporal cluster in the null distribution exceeds the observed sum t-value in the cluster of interest.

## Spatial coefficient analysis

To test the spatial evolution of phonetic representations over time, we estimated beta coefficients by fitting linear regression (i.e. ridge regression with no regularisation, so that the coefficients were interpretable). We did this both in a decoding approach (predicting a single stimulus property from multivariate sensor activity) and in an encoding approach (predicting a single sensor response from multivariate stimulus properties). Because our goal was to estimate the model weights rather than to predict out of sample data, we fit the model on all data (i.e. no cross validation). We confirmed that the beta coefficients from the decoding model and encoding model yielded the same result and continued with the decoding coefficients for conceptual simplicity.

We averaged the coefficients across participants and took the root mean square. Then, to compute the trajectories, we created two orthogonal sensor masks. First, a y-coordinate mask whereby the posterior sensors were coded as '0' and anterior sensors were coded as '1'. Second, a x-coordinate mask, whereby the left sensors were coded as '0' and the right sensors as '1'. These masks were normalised to have a norm equal to 1. The coefficients were projected onto these masks at

each time sample, to yield a cosine distance over time, within the x-y coordinate plane.

We computed a metric of trajectory structure which was a weighted combination of range of movement ($m$, maximum cosine distance minus minimum), smoothness ($s$, mean absolute step size at each time sample) and variance ($v$, standard deviation across time samples), thus:

$$\frac{m}{s}v \tag{7}$$

We compute this metric for the x co-ordinates and y co-ordinates separately, and then average the result. To evaluate whether a measure was statistically significant, we generated a surrogate 'null' dataset. For each participant, we ran the decoding analysis with shuffled feature labels. This provided an estimate of chance decoding, which we could use to generate null processing trajectories to compare to the trajectories in our empirical data. If the trajectory emerges from chance, movements should all be centred around zero (see Supplemental Fig. 16). Whereas if the trajectory contains meaningful structure, it should significantly differ from the distribution of random trajectories (see Fig. Supplemental 15). Significance was evaluated from the proportion of instances where the observed trajectory structure metric exceeded the null distribution.

### Simulation of phoneme sequence reconstruction

To quantify how many phonemes the brain processes at once, and how it retains the order of those phonemes, we simulated responses to 4-phoneme anagrams. We estimated the spatiotemporal coefficients from the decoding model without regularisation (the same coefficients used for the trajectory analysis) for voicing and plosive manner of articulation. Simulated responses were generated by multiplying the coefficients of each feature by a noise factor, shifting the phoneme response by 100 ms for each phoneme position and summing the result. This resulted in twenty-four unique sequences without repetition (e.g., bpsz, pbzs, zspb...).

We simulated 10 responses to each unique phoneme sequence. We epoched the data around each phoneme onset and used ridge regression to reconstruct the phoneme sequence. We used cosine similarity to quantify the accuracy of the reconstruction. To evaluate statistical significance, we compared observed cosine similarity to the cosine similarity when randomly shuffling phoneme labels 10,000 times, and computed the proportion of times that the observed similarity exceeded the null distribution. We could accurately reconstruct the history of five previous phonemes significantly better than chance (1: cosine similarity 1.0, $p < 0.001$; 2: cosine similarity 0.91, $p < 0.001$; 3: cosine similarity 0.79, $p < 0.001$; 4: cosine similarity 0.38, $p < .001$; 5: cosine similarity 0.19, $p = 0.012$; 6: cosine similarity 0.1, p = .6).

### Reconstructing time since phoneme onset

We assessed whether responses encoded elapsed time since phoneme onset by applying ridge regression to phoneme-locked responses. We cropped the time dimension from 100 to 400 ms which encapsulates the time window of strongest phonetic decoding, and downsampled to 53.3 Hz. For each participant, we collapsed the trial (~50,000 phonemes) and time sample (16 timepoints equally spaced between 100–400 ms) dimensions. This yielded around 800,000 neural responses per participant, each labelled for its relative time since phoneme onset.

Using a 5-fold cross validation loop with shuffled order, we fit ridge regression using default parameters in scikit-learn. For each trial, the input features to the model were the 208-channel sensor

responses. The target features were latency since phoneme onset. Both were scaled to have zero mean and unit variance.

For each test set, we computed the model's prediction of onset latency, which is a continuous measure between 100 and 400 (shown in Fig. 4). in the absence of evidence to the contrary, the model will predict the mean latency of the test set, in this case 250 ms. The within-fold correlation across participants between the true and predicted latency since onset was highly significant ($r = 0.83$, $p < 0.001$).

### Reporting summary

Further information on research design is available in the Nature Research Reporting Summary linked to this article.

## Data availability

Source data are provided with this paper. The raw data generated in this study have been deposited in a OSF database (https://doi.org/10.17605/OSF.IO/AG3KJ). The first 21 subjects in the database correspond to the data used in this study. The subsequent 6 subjects were collected later in time, in order to create a large database for public use by the community. The stimuli we use were obtained from the Open American National Corpus.

## Code availability

We have uploaded the code to a public repository: https://github.com/kingjr/meg-masc.

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

## Acknowledgements

The authors thank Graham Flick for help with data collection. We are very grateful to William Idsardi, Arianna Zuanazzi, Joan Orpella, Pablo Ripolles-Vidal and Omri Raccah for their feedback on an earlier version of the manuscript. This project received funding from the Abu Dhabi Research Institute G1001 (AM); European Union's Horizon 2020 research and innovation program under grant agreement No 660086, the Bettencourt-Schueller Foundation, the Fondation Roger de Spoel-berch, the Philippe Foundation (JRK) and The William Orr Dingwall Dissertation Fellowship (LG).

## Author contributions

L.G.: conceptualisation; methodology; software; validation; formal analysis; investigation; data curation; writing—original draft preparation and review and editing; visualisation. J.R.K.: conceptualisation; methodology; software; supervision. A.M.: conceptualisation; writing—re- view and editing; supervision; funding acquisition. D.P.: conceptualisation; writing—review and editing; supervision; funding acquisition.

## Competing interests

The authors declare no competing interests.
