## [Peer Review File · Nature Communications]

Neural dynamics of phoneme sequences reveal position-invariant code for content and orderReviewers' Comments:

Reviewer #1:

Remarks to the Author:

Gwilliams et al. report on a phoneme decoding/mapping study, based on passive story listening in MEG.

There is much to like about this study, and for a neuro-/psycholinguistic endeavour it is admirably clearly written. In essence, I have concerns as to whether it poses the advance that it aims to be.

The limitations I see are the following:

1. since no task and behavioural read-out was employed, limited conclusion can be drawn on the psycholinguistic/behavioural correspondence of any of the neural findings.
2. a too strong dismissal of the actual (low) level of accuracy in decoding phonemes: the above-chance decoding is being treated as a fact denoting "representation" or "processing" (e.g., p. 3: "On average, phonetic features were linearly decodable for three times longer than the duration of the phoneme itself. This suggests that, at any one time, three phonemes are being processed concurrently"). What is entirely dismissed and not even mentioned in the ms., if I am not mistaken, is actual phoneme decoding accuracy. It is being mentioned in passing that accuracy is, predictably, much lower for neural than for acoustics decoding. However, especially (but not exclusively) in the case cited above (p. 3), above-chance decoding should not indicate active representation or neural "processing" (e.g., de-Wit, L. et al. (2016) *Is neuroimaging measuring information in the brain?* *Psychon. Bull. Rev.* 23, 1415–1428). Given the more theoretically inclined set-up and interpretations in this entire manuscript, such interpretational short-circuiting lowered my enthusiasm considerably.
3. limited conclusions on the neural level (beyond the mentioned dichotomy of "being represented" [above-chance classification] vs. not), as the authors chose to run all classification on sensor-level MEG data. Source transformations would have been the very obvious choice, and would be tremendously helpful in tying in the psycholinguistics/representational questions (which this ms. is concerned with) with a neurobiological/neuroanatomical level of description. The ms. would be more compelling if this effort was undertaken, as it would also simplify some of the interpretations. E.g., a focus on auditory/perisylvian regions would be possible, and/or an intercomparison amongst brain regions would foster solid interpretation.

Further comments:

1. The temporal generalisation method is used to great effect here, obviously, and is central to the tenets of this manuscript. However, given its importance, I worried that filtering details of the neural data do not receive the attention they might deserve. I am sure the author who co-invented this method (JRK) has thought long and hard about this, but the current ms. should do more to assure the reader that none of the results are dependent on (acausal?) filtering choices.
2. in places, large p-values are being used as evidence for absence of an effect. Generally, effect sizes and Bayes Factors would be the statistical tools of choice. Instead, p values are being overused. Related, it was unclear what the \hat{t} symbol denotes. It's surely not an estimated t-value? Does it denote a t^* from a permutation/resampling procedure?
3. too jargon-heavy reporting and somewhat unspecific interpretations do limit the reader's (i.e., my) potential understanding. Most clearly, this is seen in sentences like (line 69f.) "Practically speaking, that the neural responses show a diagonal rather than square generalisation pattern means the underlying activations are evolving over time: activity supporting the processing of a particular phonetic feature is either moving across cortical regions or evolving or transforming within a particular

cortical region.”

4. What worries me while reading the ms. (see also major #2 above()): Are the authors doing justice to essentially trivial effects of co-articulation when it comes to high reconstruction accuracies across time?

5. A crucial covariate for the obtained reconstruction accuracy is SPL/RMS: How much signal is there to start with? It is likely that portions with low SPL/loudness and/or low RMS in the MEG will yield lower reconstruction accuracies as well. In my mind, all models reported here should be corrected/deconfounded for SPL/RMS, or the reader should at least be assured that this is not a factor of concern here. This became particularly obvious in line 140f.: higher lexical uncertainty [→ higher ERP/evoked response?/better SNR in the MEG?] → better reconstruction accuracy?, but holds for most other parts of this ms, as reconstruction accuracy is the key dependent measure throughout. Again, see major #2 on this topic.

Reviewer #2:

Remarks to the Author:

In “Neural dynamics of phoneme sequences jointly encode invariance, content and order”, Gwilliams et al. use MEG data from participants listening to narrative stories to address how the brain processes sequences of phonemes. Using a decoding paradigm, they show that they are able to determine the phonetic content of narrative stimuli over different sliding windows, and that for any given point in time, the representations of multiple phonemes are shown to exist. These results are interesting and informative for researchers interested in perception of ongoing speech/sound sequences.

Strengths:

1. The use of narrative stories and MEG on a relatively large participant cohort
2. Their use of decoding with specific time windows to uncover the representation of sequences is novel and interesting
3. The comparison between acoustic and neural decoding methods shows how brain representations for specific speech features evolve over time, whereas acoustic features do not.
4. The authors look at how MEG spatiotemporal dynamics can be used to decode phonemes while also showing the influence of surprisal, lexical entropy, position, and duration.

Weaknesses:

1. Some of the analyses would benefit from a more detailed explanation (e.g. Figure 2).
2. Many of the claims in the Discussion need to be developed more deeply – for example, the claim of representing sequence information is not very clear if phoneme representations are completely position invariant.

Some specific suggestions for improving the manuscript are listed below, split into Major and Minor comments:

Major comments:

1. Figure 2 displays the main results from which the paper’s claims are derived. While it is a very aesthetically nice figure, it takes a while to unpack the information, which makes the claims somewhat confusing. Because each of the contours come from a distribution of t-statistics from train/test time pairs that are then overlaid, it might help to break down this figure more in Figure 1 or in an additional earlier panel in Figure 2. Either that, or a clear breakdown of how to interpret each of the contours in the text would be helpful. Related to this point, the time slices (marginals) would also be clearer if the authors were to draw dashed vertical and horizontal lines through $x=0.5$ and $y=0.2$, respectively.
2. Related to all plots showing t values, the authors should also give more information about the number of points in the analysis, degrees of freedom, etc. It’s not clear exactly over what data these

values are calculated.

3. The authors claim that (line 82-83) "although multiple phonemes are processed in parallel, any given pattern of neural activity only represents one phoneme at a time." If interpretable, it would be helpful to see what exactly these spatial patterns are, at least for some example phonetic features (e.g., Spatial Pattern A relates to fricatives, Spatial Pattern B relates to low back vowels). If, for example, it's possible to see the evolution of that spatial pattern such that part of Spatial Pattern A still exists once Spatial Pattern B has started (which is suggested by the overlapping phoneme representations found here), then this might make the other analyses more intuitive.

4. In the decoding analysis, was the output of the decoder actually the phoneme itself, or a set of phonetic features? Throughout the manuscript, the decoder is presented as being on phonemes, but then in the methods the modeled features appear to be manner/place of articulation and voicing. This is fine either way, but should be clarified.

5. With regard to the phoneme duration analysis, it is intuitive that longer phonemes would be able to be decoded for a longer period of time than shorter phonemes. I wonder, then, is it also the case that the number of phonemes that are decodable at a given test time (as in Figure 2) varies with phoneme length?

6. Line 97-99: The claim that the features of all three phoneme positions could be decoded with comparable performance (and the position-invariant encoding scheme) seems at odds with the authors' claim that their data can explain the encoding of "pets" vs. "pest" – in this case, if "ets" and "est" are treated as the same "bag of phonemes", how then would the exact order emerge? I expect based on Figure 1 that it should be something about the specific spatiotemporal patterns observed in their MEG data, but this could be made more explicit. Otherwise, it doesn't seem that the data support the ability to tell these stimuli apart.

7. In the Discussion section, it is difficult to evaluate whether the analyses presented satisfy the claims that are being made. For one, the decoder performance is always shown in terms of beta weights or t-values, which can show whether the decoder is significant or not, but does not immediately tell the reader whether the performance was actually very accurate or not. Could the authors also report the percent correct for each feature, or a confusion matrix showing which particular errors were made by their decoder?

8. Regarding the claim about these representations serving to maintain overlapping sequences without confusing the content of the signal (line 157-158), could the authors perform an explicit analysis to show how sequence information is maintained (e.g. for a specific example like "pets" vs "pest" mentioned in the manuscript)?

Minor comments:

1. Figure 1: It took me a while to understand the significance of the word offset analysis in Figure 1B. At first, my interpretation was that the phonemes closest to the start of a word were most decodable from activity close to that time period, and that also phonemes closest to the end of that same word were the most decodable, but that middle phonemes were the least decodable. However, upon rereading it, I think this is to say that the neural decoder works best for phonemes near the word start, as well as near the end of the previous word (not the current word!) If this is true, could the authors please clarify this point in the text?

2. Figure 1D: In the text (line 61), this is referred to as "square" temporal generalisation. To clarify that this refers to the shape of the plot and is not a specific technical term, I'd suggest adding something like "The square shape of the temporal generalization contour suggests..."

3. For the contours in Figure 1D, these represent 95% and 90% percentile decoding accuracy. What were the actual decoding accuracy values? I interpret this to mean the 90 and 95th percentiles of the distribution of decoding accuracies, but it is not clear what those accuracies actually were.

4. In Figure 1E, it might help to show the unity line to clarify the (lack of?) asymmetry in the train vs. test time performance contours.

5. Line 107-109: I'm not sure how to interpret this sentence: "The latency between average neural and acoustic maximum accuracy was 136 ms (SD = 13 ms) at word onset and 4 ms (SD = 13 ms) at word offset (see Figure 1B), leading to a significant difference between onset and offset phonemes averaged over phonetic features."

6. Line 127-141: For the lexical identification analysis, why not align to word offset rather than onset? Alternatively, why not align to a drop in surprisal for each particular word (that is, the point at which the identity of that word is no longer ambiguous), rather than the same fixed window across all words?
7. Line 305: Was surprisal calculated on the word level, or on the phoneme level? For Figure 3, this says "phoneme surprisal" – how is this calculated from the $P(w|C)$ if w is a word?
8. Line 323 – More explanation of the rationale behind the back-to-back ridge regression model would be helpful. What is the advantage from doing this versus comparing the predicted to the actual features in your test set using your trained model?
9. I found it slightly misleading that the stimuli were not mentioned to be synthesized until the very end of the manuscript – perhaps a short statement could be added to the results section (line 29 – "listened to four short stories spoken using synthesized voices from Mac OS text-to-speech").
10. How much data in total went into the decoding analyses? The length of each stimulus is described, but because the participants were also asked questions to gauge their attention (which is a good idea), it's not clear exactly how much training/test data was used for the models. This can help for others who might wish to repeat the study.

Reviewer #3:

Remarks to the Author:

In "Neural dynamics of phoneme sequences jointly encode invariance, content and order" Gwilliams et al decode MEG responses to reconstruct single phonemes (or their voice, place or manner of articulation) to study how phoneme sequences occurring in natural language are encoded by the brain. Their goals, as they stated them, are to investigate "how the language system (i) simultaneously processes acoustic-phonetic information of overlapping inputs; (ii) keeps track of the relative order of those inputs; and (iii) maintains information sufficiently long enough to interface with (sub)lexical representations."

The results obtained for the first goal are quite convincing and very interesting; they show that neural activity (as measure by MEG) can be used to simultaneously decode up to three phonemes at any point time. Moreover, the code for this multiplexing changes dynamically so that the neural signals for the three phonemes don't intermix. I found that result very interesting (and well-illustrated in Fig 2). A minor disagreement on wording here is that the authors state that there is at least three phonemes but they show at most three phonemes...; of course, one could argue that they also don't have access to all of the neural responses thus the lower bound but this is not clearly stated. I also find that a further investigation of this code would be enlightening (as they themselves conclude). For example, one could examine the spatial distribution of the activity using the coefficients for the 208 channels during early, middle and late stages of processing. The authors have that data and have thought about that analysis...

For goal number (ii), I don't really see that they can make a strong conclusion. It is possible that relative position of phones in a triphone sequence ABC could be decoded by simultaneously attempting to decode ABC at the onset of C but this exercise was not done in this paper. Yes – I agree the information appears to be there but it needs to be quantified (and might require further exploration of the code)

For goal number (iii), the authors show a small effect of phonemic expectations and lexical entropy on the time course of phoneme prediction. This modulation of the dynamics shows that there is an interaction between phonemic and lexical representations. Given that the effect size is very small, these investigations might require further analyses (see below).

In summary, I found the paper very interesting although difficult to read at times and with over ambitious goals stated in the introduction and (some) overreaching conclusions in the discussion. This said, the result summarized in Fig 2 on the joint coding of multiple phonemes is novel and fundamental. I made a list of points that are a mixed of more major grievances and minor comments.

1. Intro | 14-15. To me "pets" and "pest" is not the same sequence of phonemes – maybe you want to

say "the same set of phonemes...".

2. L. 18. Not sure you need the subscript i on phoneme

3. Figure 1B. Why does the scale have 0.0001 and then 150? I assume that 0.0001 is neural data and 150 for acoustic but then I am confused on what exactly is your Beta coef (as described in your methods it cannot be greater than 1). Also, these scales and labels are very small.

4. L35. What is a "one-hot encoding"? You mean one-shot encoding?

5. L40-43 – here you should specify the length of the window used in the decoding analysis. I am assuming that you are using a sliding time-window of fixed length.

6. In 1B, 1C: Is the difference in maximum prediction between P1, P2, etc significant? And to what extent do you observe this because you have different amounts of data for P1, P2, etc. Do you still get a different in prediction if you use the same amount of training and testing events for each?

7. Figure 2 is really neat!

8. Paragraph 1.5 and fig 3B: why restrict your analysis to second, third and offset?

9. Paragraph 1.6. I think that the difference in latencies at word onset vs offset is interesting and well quantified by the differences in between neural and acoustic decoding, as you have done. The differences in neural decoding for place, manner and voicing shown in Fig 3C is less convincing. There is probably something interesting there but it requires further analysis. For example, I suggest also quantifying something like a cumulative decoding. You will also have to do some control for the fact that the stimulus statistics of phonemes at beginning and end of words are not identical

10. Paragraph 10.8. It makes sense that higher entropy phonemes could be decoded later but to validate this result, you should also check to see if higher decoding accuracy is not found for other windows of analysis (From 3F – it looks like it starts before 300 ms). Also, as in point 9, I think that a cumulative analysis of decoding performance would be more appropriate.

11. I don't really understand the sentence: "For (ii), we grouped trials based on non-onset cohort entropy, and ran the analysis on all phonemes that did not occur at word onset (~4500 trials per bin, mean values of 0.03, 0.77, 2.04 and 4.48 bits)"

12. In the discussion on l. 158-159, you state: "Second, relative position is implicitly coded in the representational format of each phoneme at a given input time". But you have only shown differences in onset vs offset latency. And in fact, in 3B, you show position invariance. What exactly do you mean here?

13. Discussion l. 162-174. I don't think that you can make such strong statements. Your conclusion on position invariance is based on decoding a single phoneme. But, what if you obtained better decoding of phonemes by decoding both phonemes and diphones together, or phonemes and diphones and triphones together. There might be a invariant encoding for single phoneme that is added to an invariant decoding for diphones, etc... Moreover, you have shown invariance in terms of decoding accuracy but not in terms of invariance in the weights of your decoder.

14. Discussion l. 195-198. You are mentioning a cumulative information which reinforces my suggestions to investigate cumulative measures of decoding (see points 9 and 10).

15. Methods (Section 3.5). Isn't ridge regression define by L2 (Gaussian) regularization? What is ridge with Laplacian ? Did you mean Lasso regression?

16. Methods (Section 3.8) and as it relates back to point 3. If I understand correctly, you are choosing to show the Beta from the second Ridge regression as your goodness of fit. If you had no correlation (and no regularization and no-crossvalidation), this would be the same as calculating the Pearson correlation coefficient between the prediction and the actual value. But in your plots you show a single Beta value. I assume that this is the averaged Beta for all phonemes? Also peaks of $2e-5$ is very small! Also why are 3A and 3D $2e-5$, 3B $4e-4$ and 2B and C $1e-4$? Could you provide a better way for the reader to assess the effect size. For example, what are the Betas for the acoustic feature? (I don't think 150 is correct as mentioned above).

17. Methods (Section 3.8). I assume that the analysis was performed in 2.5 ms sliding windows (250Hz) ? This was not specified.

18. Methods (Section 3.11). I am sure that the permutation test in MNE is fine but given that the Betas are so small, it would be interesting to show us the histogram of Beta values obtained by permutation test and how it compares to the Betas observed (at least for some of the data). Also, there is both significance within subject and patterns across all subjects that also need significance

testing. It was not clear to me how you performed the statistics at these two levels.

Reviewer #4:

Remarks to the Author:

The paper by Gwilliams describes a number of phoneme-decoding analyses applied to MEG recordings of continuous speech.

Aspects of this paper that are done very well include (1) rigor in attempting to control for stimulus correlations both by considering a large number of features and using both forward and backward regression to attempt to partial out the contribution of different features (2) an attempt to understand how the representation of a phoneme is modulated by context within a word – clearly an interesting question that has not received a great deal of attention.

While the findings on phoneme surprisal and lexical entropy are potentially interesting (although the context can be improved, for example the relation with another MEG study by Brodbeck & Simon, 2018), not much new can be concluded from the preceding analyses. The fact that phoneme decoding does not generalize across time has been shown before and likely reflects the fact that responses propagate from region to region in the auditory system – a basic fact that is well established. The fact that this generalization falls below the $p < 0.05$ level at about the average duration of a phoneme also seems hard to interpret. This cutoff is arbitrary and will be affected by the SNR of the data. And of course, MEG is a very coarse measure of the neural response that pools across multiple different brain regions, each of which probably have a different time constant. What is most relevant is the time constant of the single neuron responses within the regions of auditory cortex that the brain is decoding from – who knows how this relates to what is being measured here. The results of applying the cross-decoding to the spectrogram (Fig 1D) do not make much sense. Phonemes clearly have acoustic cues that vary with their onset and offset even if their acoustic realization is modulated substantially by contextual factors. Figure 1D seems to imply that one would not be able to distinguish the phonemic content over ~ 300 millisecond window, which is clearly not true. This result makes question the interpretation of the neural data. The fact that decoding is more sustained for longer phonemes (Fig 3A) and is at least partially consistent across position (Fig 3B) is not surprising as there are features that are shared across these contexts.

Major points:

1. The fact that phonetic information is decodable up to the same duration after onset (~ 300 ms) from acoustics and neural responses somewhat casts a doubt on the claim that the dynamic evolution seen in the neural decoding is a result of signal transformation, i.e. maintaining content-temporal representation. This suggests that later phonemes carry information about previous phonemes which could be utilized by the brain. For example, what do the authors predict will happen if a phoneme is followed by silence? Would the neural response for the last phoneme follow the same diagonal dynamics?
2. The phoneme surprisal and lexical entropy effects seem potentially interesting and perhaps warrant refocusing around these findings. Three missing controls would make this more compelling:
 - (I) The same analysis applied to a spectrogram (preferably something closer to a cochlear representation – see below). This seems like an obvious and basic control.
 - (II) A similar analysis but performed by a nonlinear decoder applied to just portion of the spectrogram that spans the phoneme. A nonlinear decoder is important, since it is not a priori clear what features the brain might be using to represent phonemes, and a spectrogram is clearly a highly impoverished representation. The decoder would be trained to decode a given phoneme using only the information present in that phoneme, since otherwise the network could make a prediction based on preceding information which is the question the authors are trying to address.
 - (III) Presumably the distribution of phonemes varies as a function of word position and things like surprisal and entropy. To ensure, that these differences cannot explain the results, it would be helpful

to repeat the neural analyses while attempt to match this distribution. It would also be helpful to perform some basic acoustic analyses to ensure that there are not differences in the average spectral properties or durations of phonemes as a function of word position and surprisal / entropy.

3. The effects of word onset/offset do not appear consistent across features. For example, place is decoded earlier for offsets consistent with the surprisal results. But this does not seem true for manner or voice. Is there any reason for this? Are the effects also inconsistent for surprisal and entropy? This makes me suspect there is some acoustic confound that is not be controlled for (despite already impressive attempts by the authors to control for acoustic confounds).

4. The uncovered evidence suggests that phonemes that cause a higher entropy are maintained for a longer duration compared to their low entropy counterparts, but the authors make a conclusion that phonemes are maintained until lexical identification is accomplished. The difference between the two cases (high vs low entropy), however statistically significant, seems to be less than 30ms. This amount of extra representation time for a phoneme seems hardly sufficient for reaching the point of lexical identification, which is potentially towards the very end of the word.

5. It is unclear how the static representation in the acoustic space will behave when presented with a sequence of phonemes (something similar to Figure 2), since there will be inevitable overlap between adjacent phonemes. Does this mean that phoneme position in the word is important in its acoustic representation unlike in neural representation? Please comment?

6. Is it possible that the overwhelmingly strong decoding from acoustics is because of a trivial solution to the problem and a lower SNR acoustics would lead to a more diagonal profile?

7. According to P1 in figure 2 and line 105, phonetic features are decodable earlier at word onset than offset. This seems contrary to the finding that predictable phonemes induce earlier responses, considering that offset phonemes are likely to be more predictable than onset phonemes.

More minor points:

1. The definition of surprisal is not clear. What is the numerator in equation 1? The frequency of the actual word that has perhaps not yet been completed? Why is this a measure of phoneme surprisal?
2. The authors should give an intuitive definition of cohort entropy when it is introduced in the Results, and why they are examining this parameter.
3. The authors use a spectrogram with linear frequency spacing and I believe with no compression. This is a fairly poor model of even cochlear representations. It would be better to at least use something with logarithmic spacing and some type of compressive nonlinearity (e.g. power, logarithmic) to account for the enormous dynamic range of sound.
4. I assume word "offset" means the last phoneme in a word? And "P-2" means the second to last phoneme in a word? It would be helpful to clarify this.
5. Equations 3 & 4 are missing a minus sign.
6. Equation 4 contradicts the description. The description says they are mapping from the true features to the predictions, but the equation appears to say the opposite (though hard to be sure without the minus sign).
7. Permutation analyses are not described at all. Need to at least say what was permuted.
8. Figure 2 seems at odds with the figure caption and line 78 in terms of the distance between adjacent phonemes. It looks as if phonemes are spaced roughly twice the average phoneme length, i.e. 160 ms instead of 80 ms. For example, P3 is supposed to start at t=160 but starts after t=320. This might be a simple mislabeling of the time axis, or I might be missing something.
9. For the figures that plot overall decoding accuracy, what is being plotted? The average of all the beta weights for all features?
10. I think it would be helpful to have a small legend for Figure 1A that indicates what the colors

mean. You could also have two colors one locked to word onset and one locked to word offset, which would help clarify the next figure.

11. In figure 1B, say "mean phoneme duration" rather than "phoneme duration" since phonemes are hugely variable in duration.

12. Label 0 as phoneme onset in figure 1B.

13. Figures 3A-C don't have x-axis labels.

14. Line 78: plot -> plotted

15. Line 105: then -> than

16. Line 139: we -> were

We are extremely thankful to the four reviewers for taking the time to share their thoughtful and constructive feedback on our work.

We have substantially altered the manuscript following the reviewers' suggestions, including the following main changes in the main text:

- (1) Addition of control acoustic analyses, including applying decoding to the cochlea-gram and to data with variable levels of random noise (Figure 1C)
- (2) Simulated reconstruction of phonological sequence in order to provide an empirical estimate of how many speech sounds are processed concurrently (Figure 2B)
- (3) Generalisation analysis that assesses the similarity in neural patterns between phonemes at different locations in the word (Figure 2C)
- (4) Simplification of the main result figure (now Figure 3) with the addition of effect sizes and cumulative decoding performance
- (5) Additional analyses assessing how linear order can be reconstructed from neural responses (Figure 4A)
- (6) Additional analyses that inform the spatial evolution of phoneme processing (Figure 4B)
- (7) Provision of decoding effect sizes throughout

And the following to supplementary materials:

- (1) Breakdown of decoding into different phonetic features
- (2) Comparison of back-to-back regression decoding method and classic logistic regression decoding approaches
- (3) Overlap decoding analysis applied to cochlea-gram
- (4) Cumulative decoding applied separately to each phonetic feature
- (5) Cumulative decoding as a function of entropy and surprisal
- (6) Auditory analysis applied to different signal-to-noise ratios, demonstrating that dynamic encoding does not trivially emerge from noisy decoding conditions
- (7) Additional analyses quantifying the number of phonemes that can be concurrently decoded from a single brain response
- (8) Analysis demonstrating that overall strength of the MEG signal (RMS) does not explain systematic variability in decoding performance

Below, we paste the original comments of each reviewer and respond point-by-point.

REVIEWER 1

Gwilliams et al. report on a phoneme decoding/mapping study, based on passive story listening in MEG.

There is much to like about this study, and for a neuro-/psycholinguistic endeavour it is admirably clearly written. In essence, I have concerns as to whether it poses the advance that it aims to be.

The limitations I see are the following:

1. since no task and behavioural read-out was employed, limited conclusion can be drawn on the psycholinguistic/behavioural correspondence of any of the neural findings.

Reviewer 1 is right that the present study investigates the neural representations of phonetic features, rather than the behavioral relevance of such representations.

Given the costs of neuroimaging, probing behavior can be particularly challenging. Phonemes can be presented at a rate of ~770 per minute (based on the average phoneme duration of 78 ms), whereas if we probed behaviour once every minute, and each response takes 3 seconds, this would reduce our available trials by 20% - from 25,000 trials per hour to 20,000 trials per hour.

This issue likely explains why many neuroimaging studies of speech processing now make use of passive listening investigations [Broderick et al., 2021 (Scientific Reports), Di Liberto et al., 2019 (NeuroImage), Huth et al., 2016 (Nature)].

To address this issue, we now highlight this point in the discussion (lines 396-399):

“Our study has several shortcomings that can be improved upon in future work. First, we chose a passive listening paradigm, in order to optimise the number of trials we could obtain within a single recording session. However, this prohibits us from associating decoding performance with behaviour and task performance.”

2. a too strong dismissal of the actual (low) level of accuracy in decoding phonemes: the above-chance decoding is being treated as a fact denoting “representation” or “processing” (e.g., p. 3: “On average, phonetic features were linearly decodable for three times longer than the duration of the phoneme itself. This suggests that, at any one time, three phonemes are being processed concurrently”). What is entirely dismissed and not even mentioned in the ms., if I am not mistaken, is actual phoneme decoding accuracy. It is being mentioned in passing that accuracy is, predictably, much lower for neural than for acoustics decoding. However, especially (but not exclusively) in the case cited above (p. 3), above-chance decoding should not indicate active

representation or neural “processing” (e.g., de-Wit, L. et al. (2016) Is neuroimaging measuring information in the brain? *Psychon. Bull. Rev.* 23, 1415–1428). Given the more theoretically inclined set-up and interpretations in this entire manuscript, such interpretational short-circuiting lowered my enthusiasm considerably.

De-Wit et al’s paper argues that it is not because we can read-out information from brain activity, that this information is actually used by the brain. This issue is commonplace in cognitive neuroscience (e.g. Rik Henson 2005 “What can functional neuroimaging tell the experimental psychologist?) and has been extensively discussed and formalized with multivariate encoding and decoding of brain activity (How Does the Brain Solve Visual Object Recognition?, Di Carlo et al 2012; Representational geometry: integrating cognition, computation, and the brain, Kriegeskorte Kievit TICS 2013; King & Dehaene TICS 2014; Nasarelis et al (2011) Encoding and decoding in fMRI). The current status quo, highlighted in the latest “Adversarial Collaboration” of the Computational Cognitive Neuroscience conference is that linearly-decodable information can be read-out by any neuron, and thus form a plausible bases to formally qualify a brain representation (Ivanova et al 2021: Is it that simple? Linear mapping models in cognitive neuroscience).

However, we agree that our description and interpretation can be prone to confusion, and that special care should be given to the effect sizes and the signal-to-noise ratio.

We have thus made three changes to address this point:

- (1) we now highlight that the actual decoding performance is relatively low: L65-67: “the average significant decoding accuracy is modest, at around 51-52%, but given the number of phonemes presented to the subjects, it is strongly significant: all $p < 0.01$ ”
- (2) we now adjust all reports of decoding performance to be relative to a noise ceiling “proportion of variance explained”: e.g. L73-74: “phonetic features were decodable on average between 50-300 ms from the MEG signal and accounted for 46.2% of variance explained by the full suite of 31 features”. In particular Figure 1 now shows effects varying between 2% and 25%.
- (3) we now add a simulation to test the number of concurrent phonemes encoded, without depending upon decoding performance (section 3.13).

Overall, these novel analyses strengthen our original conclusion: on average, MEG activity contains linearly-readable information about three successive phonetic features at each time sample.

3. limited conclusions on the neural level (beyond the mentioned dichotomy of “being represented” [above-chance classification] vs. not), as the authors chose to run all classification on sensor-level MEG data. Source transformations would have been the very obvious choice, and would be tremendously helpful in tying in the

psycholinguistics/representational questions (which this ms. is concerned with) with a neurobiological/neuroanatomical level of description. The ms. would be more compelling if this effort was undertaken, as it would also simplify some of the interpretations. E.g., a focus on auditory/perisylvian regions would be possible, and/or an intercomparison amongst brain regions would foster solid interpretation.

We agree that having both a high temporal resolution and a high spatial resolution would be ideal. Unfortunately, the single trial data are too noisy to produce reliable source estimates of localised MEG data. In lieu of this, we have added topographic plots which show the evolution of (non-regularised) decoding model coefficients over time. We show the movement of coefficients in Figure 4 of the main text:

The model coefficients are shown for each feature, over time and space, in supplementary materials.

We also highlight this limitation in the discussion (L400-405):

“Second, using the non-invasive recording technique of MEG allows us to conduct long recording sessions in healthy individuals. However, the spatial resolution and the signal-to-noise ratio of MEG remain insufficient to clearly delineate the spatial underpinning of the dynamic encoding scheme. This limited signal-to-noise ratio also explains the relatively low decoding performances - peaking only 1-2% greater than chance in most cases. Future work may benefit from recording a greater number of repetitions of fewer speech sounds to allow for signal aggregation across trials.”

Further comments:

1. The temporal generalisation method is used to great effect here, obviously, and is central to the tenets of this manuscript. However, given its importance, I am worried that filtering details of the neural data do not receive the attention they might deserve. I am sure the author who co-invented this method (JRK) has thought long and hard about

this, but the current ms. should do more to assure the reader that none of the results are dependent on (acausal?) filtering choices.

Thank you for bringing up this point. We have re-run the analysis of voicing decoding on single-subject data and confirm that adjustments to the filter settings do not alter the temporal generation result. In the figure below, the same analysis is applied to data that have been low passed at 10, 20, 30 and 40 Hz.

We have added the point that the results are not dependent on filter choices to the methods.

2. in places, large p-values are being used as evidence for absence of an effect. Generally, effect sizes and Bayes Factors would be the statistical tools of choice. Instead, p values are being overused.

We agree with this comment and have thus have changed our reporting to systematically convey effect sizes in addition to p-values.

Related, it was unclear what the \hat{t} symbol denotes. It's surely not an estimated t-value? Does it denote a t^* from a permutation/resampling procedure?

Yes, we use used it to denote the observed critical t-value used in the permutation test. We have now made this explicit in the Methods section.

3. too jargon-heavy reporting and somewhat unspecific interpretations do limit the reader's (i.e., my) potential understanding. Most clearly, this is seen in sentences like (line 69f.) "Practically speaking, that the neural responses show a diagonal rather than square generalisation pattern means the underlying activations are evolving over time: activity supporting the processing of a particular phonetic feature is either moving across cortical regions or evolving or transforming within a particular cortical region."

We agree with this point and have edited the text in multiple places to avoid this issue.

4. What worries me while reading the ms. (see also major #2 above: Are the authors doing justice to essentially trivial effects of co-articulation when it comes to high reconstruction accuracies across time?)

It is possible that the long duration decoding performance is due to co-articulation, given that we also see ~300 ms decoding from the spectrogram. We now make this point explicit in the manuscript.

Importantly, however, the main finding of this section, i.e. the 80 ms evolution speed rather than the overall decoding time – is difficult to trivially relate to co-articulation. In particular, we do not observe such a phenomenon in the audio spectrogram of speech sounds [Supplementary Figure 7], included below:

5. A crucial covariate for the obtained reconstruction accuracy is SPL/RMS: How much signal is there to start with? It is likely that portions with low SPL/loudness and/or low RMS in the MEG will yield lower reconstruction accuracies as well. In my mind, all models reported here should be corrected/deconfounded for SPL/RMS, or the reader should at least be assured that this is not a factor of concern here. This became particularly obvious in line 140f.: higher lexical uncertainty [→ higher ERP/evoked response?/better SNR in the MEG?] → better reconstruction accuracy?, but holds for most other parts of this ms, as reconstruction accuracy is the key dependent measure throughout. Again, see major #2 on this topic.

This is a great point, thank you for bringing it up.

The confound of loudness was controlled in the initial stages of preprocessing, where we regress out all variance that can be explained by the auditory envelope and spectral content. We now clarified our methods when we describe the TRF residual procedure: “All subsequent analyses were applied to the residuals of this model, and so can be interpreted in light of the confounds of loudness and pitch being removed.”.

The issue of MEG signal strength is an important one. It is certainly possible that features which generate larger response amplitudes lead to more robust encoding. To test whether this is a point of concern, we added RMS of each phoneme epoch to our decoding model. We found that this did not account for variance in the MEG decoding results, and it did not significantly modulate how well we could decode other features in the model, either:

We have added this analysis to supplementary results, and discuss in lines 759-773 that this lack of effect may be due to our stimuli: continuous speech does not elicit clear evoked responses, and so single-trial variability in signal strength is negligible: “One potential confound in decoding performance is the overall signal strength (e.g. magnitude of the MEG response). It is possible that stimulus features which lead to larger responses will, in turn, aid better decodability of the other features encoded in that response. To test whether this was the case, we computed the root mean square (RMS) of the sensor data and attempted to decode this from the z-scored MEG

responses, from 200 ms to 1000 after to each phoneme onset. For this, we used a Ridge regression decoder with Spearman R as the performance metric. First, we found that overall signal strength did not show the temporal dynamics that were elicited by any of our stimulus features of interest (see below figure). Second, we fit a mixed effects regression model between the single-trial RMS and decoding accuracy, for phonation, manner, and place of articulation. We modelled random slopes per subject and repetition. There was no significant relationship for any of the three features p 's $> .3$. Overall this suggests that the features we show to interact with phonetic encoding strength (e.g. surprisal and entropy) cannot be explained by the global strength of the signal. We believe the lack of effect may be due to our stimuli: continuous speech does not elicit clear evoked responses, and so single-trial variability in signal strength is negligible”.

In addition, the majority of our models are trained on all trials, meaning that our training procedure should not be affected by differences in signal strength. For the cases where we test model performance on different subsets of trials, we primarily interpret latency shifts in the temporal generalization analysis as opposed to absolute or peak modulations in decoding performance (e.g. Figure 5).

REVIEWER 2

In “Neural dynamics of phoneme sequences jointly encode invariance, content and order”, Gwilliams et al. use MEG data from participants listening to narrative stories to address how the brain processes sequences of phonemes. Using a decoding paradigm, they show that they are able to determine the phonetic content of narrative stimuli over different sliding windows, and that for any given point in time, the representations of multiple phonemes are shown to exist. These results are interesting and informative for researchers interested in perception of ongoing speech/sound sequences.

Strengths:

1. The use of narrative stories and MEG on a relatively large participant cohort
2. Their use of decoding with specific time windows to uncover the representation of sequences is novel and interesting
3. The comparison between acoustic and neural decoding methods shows how brain representations for specific speech features evolve over time, whereas acoustic features do not.
4. The authors look at how MEG spatiotemporal dynamics can be used to decode phonemes while also showing the influence of surprisal, lexical entropy, position, and duration.

Weaknesses:

1. Some of the analyses would benefit from a more detailed explanation (e.g. Figure 2).

2. Many of the claims in the Discussion need to be developed more deeply – for example, the claim of representing sequence information is not very clear if phoneme representations are completely position invariant.

Some specific suggestions for improving the manuscript are listed below, split into Major and Minor comments:

Major comments:

1. Figure 2 displays the main results from which the paper's claims are derived. While it is a very aesthetically nice figure, it takes a while to unpack the information, which makes the claims somewhat confusing. Because each of the contours come from a distribution of t-statistics from train/test time pairs that are then overlaid, it might help to break down this figure more in Figure 1 or in an additional earlier panel in Figure 2. Either that, or a clear breakdown of how to interpret each of the contours in the text would be helpful. Related to this point, the time slices (marginals) would also be clearer if the authors were to draw dashed vertical and horizontal lines through $x=0.5$ and $y=0.2$, respectively.

Thank you for highlighting this issue. We have now broken down this figure to remove the marginals. We also add a preceding figure which introduces the idea of phoneme decoding in different positions, which we hope will help readers more easily digest the information.

Figure 2 now looks like this:

And Figure 3 now looks like this:

2. Related to all plots showing t values, the authors should also give more information about the number of points in the analysis, degrees of freedom, etc. It's not clear exactly over what data these values are calculated.

We have exchanged t-values for proportion of variance explained in order to better represent raw effect sizes. Where applicable in the manuscript we more fully describe the derivation of test statistics: e.g. lines 76-77: "Performance averaged across features was statistically greater than chance, as confirmed with a temporal permutation cluster test, based upon the a one-sample t-test (df=20) applied at each time-sample ($p < .001$; critical t averaged over time = 3.61)."

3. The authors claim that (line 82-83) "although multiple phonemes are processed in parallel, any given pattern of neural activity only represents one phoneme at a time." If interpretable, it would be helpful to see what exactly these spatial patterns are, at least for some example phonetic features (e.g., Spatial Pattern A relates to fricatives, Spatial Pattern B relates to low back vowels). If, for example, it's possible to see the evolution of that spatial pattern such that part of Spatial Pattern A still exists once Spatial Pattern B has started (which is suggested by the overlapping phoneme representations found here), then this might make the other analyses more intuitive.

Thank you for this suggestion. We have added a figure both to the main text and supplementary which shows this information. Unfortunately the signal-to-noise ratio and the spatial resolution of MEG does not allow us to make strong claims about the differences between patterns upholding different phonetic features. However, we do quantify the evolution of the topographic trajectory over time: figure 4, replicated here:

4. In the decoding analysis, was the output of the decoder actually the phoneme itself, or a set of phonetic features? Throughout the manuscript, the decoder is presented as being on phonemes, but then in the methods the modeled features appear to be manner/place of articulation and voicing. This is fine either way, but should be clarified.

Thank you for pointing this out. All analyses are applied on the phonetic features, but some statistical tests are applied to the decoding averaged across features. We have now made this point explicit in the manuscript and hope this clears up confusion: “As an important theoretical point, we are decoding features of each phoneme rather than decoding phoneme categories per se. However, when decoding phoneme categories instead of features, we observe very similar results (see supplementary Figure 19).”

We have also added a supplementary analysis in section “Testing granularity of representation” where we test different formats of representation (e.g. acoustics vs. phonetic features vs. phoneme categories). The results are comparable between the two (Supplementary Figure 19), suggesting that they are too correlated in natural speech for us to distinguish them. This figure is replicated below for convenience:

5. With regard to the phoneme duration analysis, it is intuitive that longer phonemes would be able to be decoded for a longer period of time than shorter phonemes. I wonder, then, is it also the case that the number of phonemes that are decodable at a given test time (as in Figure 2) varies with phoneme length?

That is an interesting point. If longer phonemes and shorter phonemes were decodable for the same amount of time (e.g. 300 ms), this would actually result in different numbers of phonemes being decodable at a given test time depending on their length. For instance, 6 phonemes of duration 50ms would be decodable and only 2 phonemes of length 150 ms would be decodable. Instead, we observe that phoneme duration

modulates overall decoding time (the diagonal axis). This means that the number of decodable phonemes at a given test time remains stable, regardless of duration.

Furthermore, we find that phoneme duration also modulates the speed with which the representation evolves (horizontal axis: longer phonemes evolve slower than shorter phonemes (Figure 5A). This means that, regardless of phoneme duration, representational overlap is minimized.

So, overall, this adaption in processing dynamics as a function of phoneme duration means that 3 consecutive phonemes will always be encoded in neural activity. This is described in detail in section 1.9.

6. Line 97-99: The claim that the features of all three phoneme positions could be decoded with comparable performance (and the position-invariant encoding scheme) seems at odds with the authors' claim that their data can explain the encoding of "pets" vs. "pest" – in this case, if "ets" and "est" are treated as the same "bag of phonemes", how then would the exact order emerge? I expect based on Figure 1 that it should be something about the specific spatiotemporal patterns observed in their MEG data, but this could be made more explicit. Otherwise, it doesn't seem that the data support the ability to tell these stimuli apart.

Thank you for bringing this potential confusion to our attention. Indeed, position invariance can intuitively be at odds with an order embedding. These two notions, however, are fully compatible: each phonetic feature is represented by a sequence of neural responses: A->B->C. This sequence allows an order embedding: when phonetic feature $p_{\{t\}}$ is represented in population B, $p_{\{t+1\}}$ is represented in population A. The linear readout of A and B can thus identify whether $p_{\{t\}}$ precedes or succeeds $p_{\{t+1\}}$.

Our analyses suggest that each brain pattern and its associated neural assemblies (A,B and C in this example) code for phonetic features independently of their position within a word: i.e. the sound <s> in <pets> and <pest> is represented in the same neural assemblies $A_{\{s\}}$, $B_{\{s\}}$ and $C_{\{s\}}$ (although a different moment relative to word onset).

To clarify this issue, we have now added an explicit reconstruction of phoneme order from neural responses (Figure 4A, replicated above). We have also added an explicit interpretation of this result to the discussion L332-335 "The encoding of phoneme content regardless of order, and phoneme order regardless of content, is what allows the brain to keep track of the order of speech sounds, i.e. to know that you were asked to teach and not cheat, or that you are eating melons and not lemons."

7. In the Discussion section, it is difficult to evaluate whether the analyses presented satisfy the claims that are being made. For one, the decoder performance is always shown in terms of beta weights or t-values, which can show whether the decoder is significant or not, but does not immediately tell the reader whether the performance was

actually very accurate or not. Could the authors also report the percent correct for each feature, or a confusion matrix showing which particular errors were made by their decoder?

We have replaced beta weights and t-values with proportion of variance explained. We have also added a description of the raw decoding scores to the results. In addition, we make it explicit in the discussion that the decoding effect sizes we observe are a limitation of the current study:

Line 402-405: “The limited signal-to-noise ratio explains the relatively low decoding performances - peaking only 1-2% greater than chance in most cases. Future work may benefit from recording a greater number of repetitions of fewer speech sounds to allow for signal aggregation across trials.”

8. Regarding the claim about these representations serving to maintain overlapping sequences without confusing the content of the signal (line 157-158), could the authors perform an explicit analysis to show how sequence information is maintained (e.g. for a specific example like “pets” vs “pest” mentioned in the manuscript)?

Unfortunately, there are not enough repetitions of such words in the stories to perform this exact analysis. To overcome this limitation, we now use the decoding model coefficients to simulate responses to 6-phoneme anagrams. Based on this simulation we show that the phoneme sequence can be reconstructed with the correct content and phoneme order for up to a 4-phoneme history (Figure 2B, replicated below):

Minor comments:

1. Figure 1: It took me a while to understand the significance of the word offset analysis in Figure 1B. At first, my interpretation was that the phonemes closest to

the start of a word were most decodable from activity close to that time period, and that also phonemes closest to the end of that same word were the most decodable, but that middle phonemes were the least decodable. However, upon rereading it, I think this is to say that the neural decoder works best for phonemes near the word start, as well as near the end of the previous word (not the current word!) If this is true, could the authors please clarify this point in the text?

We agree that without explanation this is confusing. The reduction in decoding performance as a function of the distance from word onset and word offset is an expected artifact caused by the variability in word length. We have now added an explanation of this to the results: “Note that the number of trials available at test time decreases as we test phoneme positions further from word boundaries, due to differences in word length. P1 and P-1 are defined for 7335 trials per subject per repetition, P2 and P-2 for 7332 trials, P3 and P-3 for 5302 trials, P4 and P-4 for 3009 trials, P5 and P-5 for 1744 trials.”

This means that, as you correctly observe, decoding accuracy is highest at the first and last phoneme of the (current) word because the maximum number of trials are available to be included in the test set of the decoder. To ensure that this is not a confound in our results, we have repeated the analysis on trial counts equalized to the fourth phoneme (3009 trials per position). Our main findings are replicated, and, as expected, decoding performance is equal across positions in this circumstance:

2. Figure 1D: In the text (line 61), this is referred to as “square” temporal generalisation. To clarify that this refers to the shape of the plot and is not a specific technical term, I’d suggest adding something like “The square shape of the temporal generalization contour suggests...”

We have made this change, thank you.

3. For the contours in Figure 1D, these represent 95% and 90% percentile decoding accuracy. What were the actual decoding accuracy values? I interpret this to mean the 90 and 95th percentiles of the distribution of decoding accuracies, but it is not clear what those accuracies actually were.

Thank you for pointing this out. We have now changed the percentile decoding accuracy contours for statistical threshold (t values of 3.5 and 4.0). We make a note elsewhere in the manuscript that the actual decoding accuracies are quite low, on the order of a few percentage points above chance level, and emphasise this is a limitation of the current study.

4. In Figure 1E, it might help to show the unity line to clarify the (lack of?) asymmetry in the train vs. test time performance contours.

We have now adjusted the figure so this is no longer a concern.

5. Line 107-109: I’m not sure how to interpret this sentence: “The latency between average neural and acoustic maximum accuracy was 136 ms (SD = 13 ms) at word onset and 4 ms (SD = 13 ms) at word offset (see Figure 1B), leading to a significant difference between onset and offset phonemes averaged over phonetic features.”

We agree that our original wording was confusing. We have now updated the text: “When averaging decoding performance across phonetic features, the lag between neural and acoustic maximum accuracy was 136 ms (SD = 13 ms) at word onset, which reduced to 4 ms (SD = 13 ms) at word offset. This average onset/offset latency difference was statistically significant”

6. Line 127-141: For the lexical identification analysis, why not align to word offset rather than onset? Alternatively, why not align to a drop in surprisal for each particular word (that is, the point at which the identity of that word is no longer ambiguous), rather than the same fixed window across all words?

The analysis is being applied to all mid-word phonemes — i.e. neither at word onset or word offset. We did this to control for the fact word offsets have high lexical certainty and word onsets have low lexical certainty. The median split on surprisal and on entropy is essentially testing what you suggest — decodability of phonetic features as a function of ambiguity. Unfortunately, the disambiguation point of most of our words occurs at the onset of the subsequent word (what would be “silence” if the word was presented in isolation). It would be a very interesting future direction to apply our analysis to words where the disambiguation point is word-internal to get a better handle on this process. We have added this point to the discussion.

7. Line 305: Was surprisal calculated on the word level, or on the phoneme level? For Figure 3, this says “phoneme surprisal” – how is this calculated from the $P(w|C)$ if w is a word?

Our apologies, this was an error in the equation. Surprisal was calculated at the phoneme level: $P(p | C)$.

8. Line 323 – More explanation of the rationale behind the back-to-back ridge regression model would be helpful. What is the advantage from doing this versus comparing the predicted to the actual features in your test set using your trained model?

The purpose was to de-confound the contribution of the different features in our model. The main advantage of B2B, as compared to standard encoding model, is to combine all MEG channels into a single “virtual sensor” that we can analyze with a standard encoding model. The advantage of this approach is to boost signal-to-noise ratio, by gathering all the little bits of information scattered across MEG channels into a single dimension. We now clarify this issue in the text L56-59: “B2B was chosen to control for the co-variance between features in the multivariate analysis while optimizing the linear combination of MEG channels to detect the encoding of information even in low signal-to-noise circumstances”.

We have also added a direct comparison of decoding timecourses for Logistic regression and back-to-back ridge regression to the supplementary materials:

9. I found it slightly misleading that the stimuli were not mentioned to be synthesized until the very end of the manuscript – perhaps a short statement could be added to the results section (line 29 “–listened to four short stories spoken using synthesized voices from Mac OS text-to-speech”).

We have clarified our use of synthesised voices in the first paragraph of the results section L36-37: “21 participants listened to four short stories recorded using synthesised voices from Mac OS text-to-speech”.

10. How much data in total went into the decoding analyses? The length of each stimulus is described, but because the participants were also asked questions to gauge their attention (which is a good idea), it’s not clear exactly how much training/test data was used for the models. This can help for others who might wish to repeat the study.

Thank you for the suggestion. We have added this information to the results section L38-39: “Each subject completed two one-hour recording sessions, yielding brain responses to 50,518 phonemes, 13,798 words and 1,108 sentences per subject”.

Reviewer #3 (Remarks to the Author):

In “Neural dynamics of phoneme sequences jointly encode invariance, content and order” Gwilliams et al decode MEG responses to reconstruct single phonemes (or their voice, place or manner of articulation) to study how phoneme sequences occurring in natural language are encoded by the brain. Their goals, as they stated them, are to investigate “how the language system (i) simultaneously processes acoustic-phonetic information of overlapping inputs; (ii) keeps track of the relative order of those inputs; and (iii) maintains information sufficiently long enough to interface with (sub)lexical representations.”

The results obtained for the first goal are quite convincing and very interesting; they show that neural activity (as measure by MEG) can be used to simultaneously decode up to three phonemes at any point time. Moreover, the code for this multiplexing changes dynamically so that the neural signals for the three phonemes don’t intermix. I found that result very interesting (and well-illustrated in Fig 2). A minor disagreement on wording here is that the authors state that there is at least three phonemes but they show at most three phonemes...; of course, one could argue that they also don’t have access to all of the neural responses thus the lower bound but this is not clearly stated. I also find that a further investigation of this code would be enlightening (as they themselves conclude). For example, one could examine the spatial distribution of the activity using the coefficients for the 208 channels during early, middle and late stages of processing. The authors have that data and have thought about that analysis...

Thank you for this suggestion. We have now added an explicit simulation to test how many phonemes are present in the neural signal without depending upon decoding performance. We state that our results represent a lower bound estimate.

We have also added an analysis which explores the evolution of the sensor-level decoding coefficients over time, replicated here for simplicity:

In addition, we have also added figures to the supplementary materials which show the evolution of model coefficients over time for each of the 31 speech features analysed in this study.

For goal number (ii), I don't really see that they can make a strong conclusion. It is possible that relative position of phones in a triphone sequence ABC could be decoded by simultaneously attempting to decode ABC at the onset of C but this exercise was not done in this paper. Yes – I agree the information appears to be there but it needs to be quantified (and might require further exploration of the code)

We have now added the analysis you suggest: decoding the features of phonemes A, B, C from the neural responses time-locked to the onset of phoneme C (left panel of the below figure). For completeness, we also ran the complimentary analysis – decoding the features of phonemes A, B, C from the neural responses time-locked to the onset of phoneme A (right panel of the below figure).

The figure below is also included in supplementary materials. The analysis confirms that the manner and phonation features of phonemes ABC can be decoded both from the beginning of the sequence (A) and from the end of the sequence (C).

For goal number (iii), the authors show a small effect of phonemic expectations and lexical entropy on the time course of phoneme prediction. This modulation of the dynamics shows that there is an interaction between phonemic and lexical representations. Given that the effect size is very small, these investigations might require further analyses (see below).

In summary, I found the paper very interesting although difficult to read at times and with over ambitious goals stated in the introduction and (some) overreaching conclusions in the discussion. This said, the result summarized in Fig 2 on the joint coding of multiple phonemes is novel and fundamental. I made a list of points that are a mixed of more major grievances and minor comments.

1. Intro | 14-15. To me “pets” and “pest” is not the same sequence of phonemes – maybe you want to say “the same set of phonemes...”.

Thank you, we have made this change.

2. L. 18. Not sure you need the subscript i on phoneme

Thank you, we have removed it.

3. Figure 1B. Why does the scale have 0.0001 and then 150? I assume that 0.0001 is neural data and 150 for acoustic but then I am confused on what exactly is your Beta coef (as described in your methods it cannot be greater than 1). Also, these scales and labels are very small.

The 150 refers to the acoustic and 0.0001 to the neural data. The beta coefficients are what map the true suite of linguistic and statistical features to the model's predicted values of linguistic and statistical features. That the values are so much less in the neural data is because the neural model generates more noisy predictions of the stimulus features than the acoustic model.

We no longer present the results in terms of beta coefficients. Instead, we report results relative to a noise ceiling, which allows the neural and acoustic analysis to be directly compared.

4. L35. What is a “one-hot encoding”? You mean one-shot encoding?

One-hot encoding is a binary coding of multi-class features. So, each level of “manner” for example is coded as a separate variable — one for “fricative”, “nasal”, “plosive” etc.

5. L40-43 – here you should specify the length of the window used in the decoding analysis. I am assuming that you are using a sliding time-window of fixed length.

We do not employ a sliding window. We have now made this explicit in the results: “This was applied at each time sample independently” and methods: “The model was applied independently at each of 201 time-points in the epoch. We did not use a sliding window”.

6. In 1B, 1C: Is the difference in maximum prediction between P1, P2, etc significant? And to what extent do you observe this because you have different amounts of data for P1, P2, etc. Do you still get a different in prediction if you use the same amount of training and testing events for each?

We have re-run the analysis matching the number of trials to the fourth phoneme position (P4, P-4: 3009 trials). The results are included above in response to review 2 and in supplementary materials. Once the number of trials is equal across positions, there is no significant difference in maximum prediction accuracy.

7. Figure 2 is really neat!

Thank you!

8. Paragraph 1.5 and fig 3B: why restrict your analysis to second, third and offset?

This was done because there are an equal number of trials available at these phoneme positions. We have now also added second-to-last to show that the results do not vary as a function of phoneme distance (Figure 2C, replicated below):

C
9. Paragraph 1.6. I think that the difference in latencies at word onset vs offset is interesting and well quantified by the differences in between neural and acoustic decoding, as you have done. The differences in neural decoding for place, manner and voicing shown in Fig 3C is less convincing. There is probably something interesting there but it requires further analysis. For example, I suggest also quantifying something like a cumulative decoding. You will also have to do some control for the fact that the stimulus statistics of phonemes at beginning and end of words are not identical

We agree that breaking down the word onset/offset effect into the different phonetic features is not a very convincing result. And, indeed, we believe a more parsimonious explanation of the word boundary effect is the varying stimulus statistics at the beginning and end of words – as you point out. We find that the difference in phonetic decoding at word boundaries is better modelled by surprisal and entropy L259-275: “Is this latency shift purely lexical in nature, or could statistical properties of speech sounds at word boundaries explain the result? In general, and in our speech stimuli, phonemes at the beginning of words are less predictable than phonemes at the end of words. [...] This result suggests that the brain initiates phonetic processing earlier when the phoneme identity is more certain. This result at least partially, and potentially fully, explains the latency shift at word boundaries”.

In line with this, we have also de-emphasised the onset/offset distinction in the figures.

Your suggestion for cumulative decoding is addressed below.

10. Paragraph 10.8. It makes sense that higher entropy phonemes could be decoded later but to validate this result, you should also check to see if higher decoding accuracy is not found for other windows of analysis (From 3F – it looks like it starts before 300

ms). Also, as in point 9, I think that a cumulative analysis of decoding performance would be more appropriate.

Thank you for this suggestion. When we run our analysis over a larger timewindow, we indeed observe significant effects starting from an earlier period of time. From the results section: “When we expand the analysis window to the entire 0-500 ms epoch with a lower cluster-forming threshold, we find that lexical entropy significantly modulates phonetic decoding from 200-420 ms $p = .016$, $t = -2.61$).”

As you suggested, we have also now run cumulative decoding analyses on the data split into median surprisal and entropy. We think that the non-cumulative decoding curves may be easier for the reader to interpret, so we have put these into the supplementary figures. As would be expected, decoding for phonemes with low surprisal begin to ramp earlier, and do so faster, then high surprisal phonemes. They also reach a higher asymptote. Because the main effect of entropy is later in time, the cumulative decoding visualization does not as clearly show the dynamics of temporal sustain.

11. I don't really understand the sentence: “For (ii), we grouped trials based on non-onset cohort entropy, and ran the analysis on all phonemes that did not occur at word onset (~4500 trials per bin, mean values of 0.03, 0.77, 2.04 and 4.48 bits)”

We have rephrased: “we grouped trials based on binned cohort entropy, and ran the analysis on all phonemes that did *not* occur at word onset”

12. In the discussion on l. 158-159, you state: “Second, relative position is implicitly coded in the representational format of each phoneme at a given input time”. But you

have only shown differences in onset vs offset latency. And in fact, in 3B, you show position invariance. What exactly do you mean here?

Thank you for pointing this out. What we mean is that it is possible to decode the amount of elapsed time since phoneme onset. We have now added an explicit section in the results and added Figure panel 4A to show this result:

13. Discussion I. 162-174. I don't think that you can make such strong statements. Your conclusion on position invariance is based on decoding a single phoneme. But, what if you obtained better decoding of phonemes by decoding both phonemes and diphones together, or phonemes and diphones and triphones together. There might be a invariant encoding for single phoneme that is added to an invariant decoding for diphones, etc... Moreover, you have shown invariance in terms of decoding accuracy but not in terms of invariance in the weights of your decoder.

Thank you for pointing this out. We believe that our results are clear evidence that there exists a position-invariant code for phonetic content. However, this does not preclude the existence of other coding schemes in parallel. We have made this point clear in the results and discussion sections: e.g. L347-349 "While it is possible that multiple representational systems co-exist, our results support that at least one of those encoding schemes is context-independent, which encodes content regardless of lexical edges."

To your second point, we have now added an explicit test of invariance of the weights of the decoder by training on the responses to a phonetic feature at one phoneme position

(e.g. onset) and evaluating decoding performance at other phoneme positions (e.g. second, third, last, second-to-last). This has been added to Figure 2C, replicated below:

14. Discussion I. 195-198. You are mentioning a cumulative information which reinforces my suggestions to investigate cumulative measures of decoding (see points 9 and 10).

This has now been added in response to your previous points.

15. Methods (Section 3.5). Isn't ridge regression defined by L2 (Gaussian) regularization? What is ridge with Laplacian? Did you mean Lasso regression?

This was a mistake. We used ridge regression with L2 regularisation. Not laplacian.

16. Methods (Section 3.8) and as it relates back to point 3. If I understand correctly, you are choosing to show the Beta from the second Ridge regression as your goodness of fit. If you had no correlation (and no regularization and no-crossvalidation), this would be the same as calculating the Pearson correlation coefficient between the prediction and the actual value. But in your plots you show a single Beta value. I assume that this is the averaged Beta for all phonemes? Also peaks of $2e-5$ is very small! Also why are 3A and 3D $2e-5$, 3B $4e-4$ and 2B and C $1e-4$? Could you provide a better way for the reader to assess the effect size. For example, what are the Betas for the acoustic feature? (I don't think 150 is correct as mentioned above).

Yes that is correct, and yes we are showing the average beta values across features. We have now made this explicit in the results section. We have now changed the effect sizes to proportion of variance explained, which also allows for a more straightforward comparison across the neural and acoustic analysis.

17. Methods (Section 3.8). I assume that the analysis was performed in 2.5 ms sliding windows (250Hz) ? This was not specified.

There was no sliding window. We first downsample the data to 250 Hz and then apply the analysis on each time sample independently. We now clarify this in the methods.

18. Methods (Section 3.11). I am sure that the permutation test in MNE is fine but given that the Betas are so small, it would be interesting to show us the histogram of Beta values obtained by permutation test and how it compares to the Betas observed (at least for some of the data). Also, there is both significance within subject and patterns across all subjects that also need significance testing. It was not clear to me how you performed the statistics at these two levels.

Thank you for the suggestion. We have now added an explicit “null” analysis, whereby we shuffle phonetic labels and re-run the decoding on this shuffled data, for each subject. Our statistical tests are applied to this set of null decoding performance where appropriate. This approach is explained in the methods section: L606-608 “To evaluate whether a measure was statistically significant, we generated a surrogate ‘null’ dataset. For each subject, we ran the decoding analysis with shuffled feature labels. This provided an estimate of chance decoding.”

Reviewer #4 (Remarks to the Author):

The paper by Gwilliams describes a number of phoneme-decoding analyses applied to MEG recordings of continuous speech.

Aspects of this paper that are done very well include (1) rigor in attempting to control for stimulus correlations both by considering a large number of features and using both forward and backward regression to attempt to partial out the contribution of different features (2) an attempt to understand how the representation of a phoneme is modulated by context within a word – clearly an interesting question that has not received a great deal of attention.

While the findings on phoneme surprisal and lexical entropy are potentially interesting (although the context can be improved, for example the relation with another MEG study by Brodbeck & Simon, 2018), not much new can be concluded from the preceding analyses. The fact that phoneme decoding does not generalize across time has been shown before and likely reflects the fact that responses propagate from region to region in the auditory system – a basic fact that is well established. The fact that this generalization falls below the $p < 0.05$ level at about the average duration of a phoneme also seems hard to interpret. This cutoff is arbitrary and will be affected by the SNR of the data. And of course, MEG is a very coarse measure of the neural response that pools across multiple different brain regions, each of which probably have a different

time constant. What is most relevant is the time constant of the single neuron responses within the regions of auditory cortex that the brain is decoding from – who knows how this relates to what is being measured here. The results of applying the cross-decoding to the spectrogram (Fig 1D) do not make much sense. Phonemes clearly have acoustic cues that vary with their onset and offset even if their acoustic realization is modulated substantially by contextual factors. Figure 1D seems to imply that one would not be able to distinguish the phonemic content over ~300 millisecond window, which is clearly not true. This result makes question the interpretation of the neural data. The fact that decoding is more sustained for longer phonemes (Fig 3A) and is at least partially consistent across position (Fig 3B) is not surprising as there are features that are shared across these contexts.

Major points:

1. The fact that phonetic information is decodable up to the same duration after onset (~300ms) from acoustics and neural responses somewhat casts a doubt on the claim that the dynamic evolution seen in the neural decoding is a result of signal transformation, i.e. maintaining content-temporal representation. This suggests that later phonemes carry information about previous phonemes which could be utilized by the brain. For example, what do the authors predict will happen if a phoneme is followed by silence? Would the neural response for the last phoneme follow the same diagonal dynamics?

This is a great point, thank you for highlighting it. We address it with two additional analyses.

First, we have added an explicit test for context-sensitive encoding in two regards – (i) later phonemes are represented in the context of previous phonemes; (ii) earlier phonemes acquire information about later phonemes as they unfold.

To test (i), the hypothesis is that the fourth phoneme in a sequence is actually encoded as {DCBA}. We trained a classifier on responses to phoneme {D} and tried to decode the features of each of the preceding phonemes in the sequence {DCBA}. The results show that the same pattern of activity which discriminates features of phoneme {D} also serves to discriminate features of phoneme {C}, which could be indicative of a shared contextual representation. Earlier phonemes {BA} in the sequence did not yield above-chance decoding accuracy. It is unclear whether this is due to poor signal to noise ratio in the data, or whether this is actually indicative of limits on the range of contextual effects in phonological processing. This is an interesting avenue for future targeted research.

To test (ii), we trained a classifier on responses to phoneme {A} and then tried to decode the features of {A} from responses to each phoneme A, B, C and D. The pattern of activity which encodes the features of phoneme {A} did not yield significant decoding performance in the responses to subsequent phonemes in the sequence.

Again, this could be due to poor signal to noise in our data, or it is potentially indicative that phoneme representations do not acquire the representation of subsequent sounds as the representation evolves over time.

Overall, without future targeted research, it is hard for us to definitively infer the existence and nature of context-specific encoding of phoneme sequences. But, these preliminary results do raise the possibility that there exists *both* a context sensitive and context insensitive encoding of phoneme sequences. We state this in the discussion, thus: L347-349 “While it is possible that multiple representational systems co-exist, our results support the existence of context-independent encoding, which encodes duration since phoneme onset, regardless of lexical edges”.

2. The phoneme surprisal and lexical entropy effects seem potentially interesting and perhaps warrant refocusing around these findings. Three missing controls would make this more compelling:

- (i) The same analysis applied to a spectrogram (preferably something closer to a cochlear representation – see below). This seems like an obvious and basic control.

We agree that it is a good idea to test for statistical features in the spectrogram, and have now added decoding of information theoretic measures from the acoustic signal to our analyses, to Figure 1, replicated below. Statistical features do have acoustic correlates, but (i) they explain much less variance in the acoustic signal than the auditory signal (ii) the strength of statistical decoding in the acoustic signal does not correlate with the strength of decoding in the neural signal (iii) this does not explain the interaction between expectancy/uncertainty and the temporal dynamics of the phonetic decoding we observe.

We have added this analysis to the results section: “there was no significant correlation between the decoding performance of statistical structure and phoneme position across the auditory and neural analysis (Spearman $r = .13$; $p = .41$), and these features

accounted for significantly more variance in the MEG as compared to the auditory analysis ($t = 2.82$; $p = .012$). Overall this analysis suggests that phonetic and statistical features of speech have correlates in the acoustic signal, but that the decoding of higher order structure from neural responses is not explained by the acoustic input alone.”

(II) A similar analysis but performed by a nonlinear decoder applied to just portion of the spectrogram that spans the phoneme. A nonlinear decoder is important, since it is not a priori clear what features the brain might be using to represent phonemes, and a spectrogram is clearly a highly impoverished representation. The decoder would be trained to decode a given phoneme using only the information present in that phoneme,

since otherwise the network could make a prediction based on preceding information which is the question the authors are trying to address.

We think that this is a little out of the scope of our goals. Given the large number of non-linear decoders we could choose from, especially with the rise of speech deep learning algorithms, it would be difficult to associate a result (or null result) directly with our neural analyses.

(III) Presumably the distribution of phonemes varies as a function of word position and things like surprisal and entropy. To ensure, that these differences cannot explain the results, it would be helpful to repeat the neural analyses while attempt to match this distribution. It would also be helpful to perform some basic acoustic analyses to ensure that there are not differences in the average spectral properties or durations of phonemes as a function of word position and surprisal / entropy.

Our model is trained on the responses to all phonemes. Only at test-time are the data sub-sampled into distinct phoneme positions. This ensures that the model weights learnt by our decoder are un-biased relative to the potential imbalances in the feature distributions relative to phoneme location you describe.

In addition, we deal with confounds between language features in our test-set by using the back-to-back regression method of evaluating decoding performance.

Furthermore, the different time course of decoding information theoretic measures versus phonetic features (Figure 1) suggests that the correlation between surprise / entropy / distance from word boundaries are not so strong as to prevent us from disassociating their contribution to the analysis model.

Finally, in order to overcome the correlation between word boundaries and surprisal/entropy in our information maintenance analysis, we subset out the boundary phonemes (i.e. do not include P1 and P-1 phonemes) in our surprisal and entropy analysis. We also checked that word length was not a better predictor of this result. This allows us to be confident that our results are due to surprisal and entropy, and not associated confounds.

Overall, we have taken substantial care to explore the role of surprisal and entropy in our data, and to ensure that these confounds do not serve to explain the claims we make regarding phonetic sequence processing more generally.

3. The effects of word onset/offset do not appear consistent across features. For example, place is decoded earlier for offsets consistent with the surprisal results. But this does not seem true for manner or voice. Is there any reason for this? Are the effects also inconsistent for surprisal and entropy? This makes me suspect there is some

acoustic confound that is not be controlled for (despite already impressive attempts by the authors to control for acoustic confounds).

All of the phonetic decoding results presented here are consisted across features; however, some features yield poorer decoding performance making their precise dynamics more difficult to assess. We have re-run the main analysis (Figure 3) separately on each phonetic feature family and added it to supplementary materials, also replicated below for convenience:

We have added an explicit breakdown of the surprisal and entropy analysis into different features to the supplementary results, also replicated here. As with the overall phonetic analysis, the information theoretic breakdown is clearest for manner of articulation, because manner is the most robustly encoded in neural activity.

4. The uncovered evidence suggests that phonemes that cause a higher entropy are maintained for a longer duration compared to their low entropy counterparts, but the

authors make a conclusion that phonemes are maintained until lexical identification is accomplished. The difference between the two cases (high vs low entropy), however statistically significant, seems to be less than 30ms. This amount of extra representation time for a phoneme seems hardly sufficient for reaching the point of lexical identification, which is potentially towards the very end of the word.

When we run the entropy analysis over the full 0-500 ms epoch time window, we see that the full range of the entropy effect actually lasts 200-420 ms. This is a reasonable time-frame within which lexical entropy could exhibit an effect, and matches the time-frame of entropy reduction in our stimulus materials remarkably well. The average entropy of the low-entropy bin was 1.19 bits, which on average occurs 333 ms into a word (SD=158 ms). The average entropy of the high-entropy bin was 4.3 bits, which on average occurs 108 ms into a word (SD=51 ms). This therefore means that the average time between a phoneme in the high entropy bin and low entropy bin was 225 ms, which matches the duration of the entropy modulation effect in our data. We have added this analysis and explanation to our results section L289-298.

We agree that this line of investigation would likely benefit from future research which manipulates word-internal lexical entropy and assess the precise consequence on how long phonetic content is maintained for. One limitation of our current stimuli, for example, is that in most cases lexical entropy does not reduce to zero until after word offset. We now discuss this point in the discussion: Line 394-396: "Future work would benefit from orthogonalizing lexical entropy and word-internal disambiguation points (i.e. when entropy reduces to zero) to more precisely quantify the temporal relationship with information maintenance."

5. It is unclear how the static representation in the acoustic space will behave when presented with a sequence of phonemes (something similar to Figure 2), since there will be inevitable overlap between adjacent phonemes. Does this mean that phoneme position in the word is important in its acoustic representation unlike in neural representation? Please comment?

Because our decoding analysis is fit on all phonemes, and then tested on subsets of phonemes in different positions, the fact that above-chance decoding exists at all for the acoustic representation suggests that there must also exist a degree of position-invariant representation in the acoustic signal, too. However, unlike neural responses which robustly encode phoneme position, this information is poorly decoded from the audio spectrogram (added to Figure 1). This result suggests that original auditory space of representation does not contain such a sequence, and that the brain actively construct it through its dynamic processing scheme.

We have now added a supplementary figure showing the equivalent of the phoneme sequence figure (now Figure 3) on the auditory signal. There is representational overlap

between the neighbouring speech sounds in the auditory analysis, which is absent in the neural analysis. Figure replicated below:

6. Is it possible that the overwhelmingly strong decoding from acoustics is because of a trivial solution to the problem and a lower SNR acoustics would lead to a more diagonal profile?

We have re-run the auditory analysis at different levels of SNR and all show a square profile. Only the maximum decodability is modulated, not the profile:

7. According to P1 in figure 2 and line 105, phonetic features are decodable earlier at word onset than offset. This seems contrary to the finding that predictable phonemes induce earlier responses, considering that offset phonemes are likely to be more predictable than onset phonemes.

Phonetic features are decodable earlier at word offset (more predictable) and decodable longer at word onset (higher entropy). The direction of this effect is therefore in line with predictability.

More minor points:

1. The definition of surprisal is not clear. What is the numerator in equation 1? The frequency of the actual word that has perhaps not yet been completed? Why is this a measure of phoneme surprisal?

We made a mistake in the equation. This has now been fixed.

2. The authors should give an intuitive definition of cohort entropy when it is introduced in the Results, and why they are examining this parameter.

Thank you for this suggestion. We have added this to the relevant section in the results: L277-279 “Our results suggest that the temporal dynamics of phonetic processing are modulated by certainty about the phoneme unit being said. Does this extend to certainty about word identity, such that phonetic representations are maintained until higher order structure can be formed?”. And L280-283 “We quantified word uncertainty using lexical cohort entropy (Gwilliams & Davis, 2021). If many words are compatible with a given phoneme sequence, then lexical entropy is high. On the contrary, if only one word is likely given a given phoneme sequence (which is more often the case towards word offset), then lexical entropy is low.”

3. The authors use a spectrogram with linear frequency spacing and I believe with no compression. This is a fairly poor model of even cochlear representations. It would be better to at least use something with logarithmic spacing and some type of compressive nonlinearity (e.g. power, logarithmic) to account for the enormous dynamic range of sound.

We have changed the acoustic analysis to a Mel spectrogram.

4. I assume word “offset” means the last phoneme in a word? And “P-2” means the second to last phoneme in a word? It would be helpful to clarify this.

We have clarified this in multiple places in the manuscript. Thank you for pointing this out.

5. Equations 3 & 4 are missing a minus sign.

Thank you for pointing that out. We have fixed it.

6. Equation 4 contradicts the description. The description says they are mapping from the true features to the predictions, but the equation appears to say the opposite (though hard to be sure without the minus sign).

Yes, we map the matrix of true features to the vector of the predicted feature.

7. Permutation analyses are not described at all. Need to at least say what was permuted.

We have added a section to the methods and also added more description to the relevant places in the results section, too.

8. Figure 2 seems at odds with the figure caption and line 78 in terms of the distance between adjacent phonemes. It looks as if phonemes are spaced roughly twice the average phoneme length, i.e. 160 ms instead of 80 ms. For example, P3 is supposed to start at $t=160$ but starts after $t=320$. This might be a simple mislabeling of the time axis, or I might be missing something.

This was a mistake, the labelling of the x-axis was incorrect. This issue has now been fixed.

9. For the figures that plot overall decoding accuracy, what is being plotted? The average of all the beta weights for all features?

Yes, this is the average over phonetic features. We have now made the distinction between single phonetic feature decoding and average phonetic feature decoding more explicit in the manuscript.

10. I think it would be helpful to have a small legend for Figure 1A that indicates what the colors mean. You could also have two colors one locked to word onset and one locked to word offset, which would help clarify the next figure.

Thank you for the suggestion, we have made this clearer in the captions.

11. In figure 1B, say “mean phoneme duration” rather than “phoneme duration” since phonemes are hugely variable in duration.

Thank you, we have fixed this.

12. Label 0 as phoneme onset in figure 1B.

Thank you, we have done this.

13. Figures 3A-C don't have x-axis labels.

Fixed

14. Line 78: plot -> plotted

15. Line 105: then -> than

16. Line 139: we -> were

All fixed, thank you.

Reviewers' Comments:

Reviewer #1:

Remarks to the Author:

I would like to thank the authors for their efforts in responding in detail and with great clarifications and new analyses to my more technical concerns, most of which have been resolved.

I value the technical level of detail with which the authors have addressed comments, and I found many if not most of my more technical concerns about potential confounds addressed satisfactorily. (But certainly not all, see esp. the chosen non-use of Bayes Factors for null effects.)

However, this is also to note that the authors' response has remained rather deflective of my and others' more substantial concerns, esp. on this study's very limited ability to elucidate speech and language perception and behaviour per se and, related, the low decoding accuracies. Both have been covered in a brief paragraph in the discussion, and I will leave it to the editors' judgement whether this is deemed sufficient.

Put most succinctly, statements like "[this study] shed new light on how the human brain combines them to probe the mental lexicon" (abstract, concluding remarks) strike me as not really addressed by the study and analyses at hand. I congratulate the authors on a technical tour de force but remain uncertain about the larger implications and relevance of this work.

Reviewer #2:

Remarks to the Author:

In this resubmission of "Neural dynamics of phoneme sequences: Position-invariant code for content and order", Gwilliams and colleagues use MEG data during natural story listening to decode the phonetic content over different sliding windows, and show that multiple phonemes (in order) can be decoded from each time sample. In the resubmission, they were responsive to my and the other reviewer's comments, and the clarity of the manuscript has improved. I appreciated the additional analyses showing the spatial evolution of phoneme processing. Although the decoding accuracy is still relatively low, they do show overall that noninvasive imaging can be used to decode phoneme sequences in natural speech. Overall, I am satisfied with their responses to the reviews.

Reviewer #3:

Remarks to the Author:

The authors have diligently addressed most of the comments and criticism from the previous review cycle. Their description of this instantaneous yet dynamical code for triphones is novel and interesting. The analyses are rigorous and complete. In the revised version, the authors have added some important controls that further validate the relevance of this dynamic representation of phonemes for linguistic processing. My major complaint: the methods are somewhat complicated, and the results section is written in a very concise form. As a result the paper is not an easy read. I think that the authors could further describe their methodology in the methods section; for e.g. by including all equations with clearly labeled variables. See also my comment below.

Major Point.

I applaud the use of effect size instead of t-values (or raw Beta coefficient values) but the calculation of the "variance explained" effect size is missing from the paper: I could not find it in the results or in the methods (or in the response to the reviewers). I suspect that this is something that might be obvious to the authors but not-so much so for readers who are trying to follow the sequence of regression analyses in B2B decoding: Using the 31 features, what variance are you trying to capture?

The variability in the decoded features using the actual features? The variability in the MEG signal using beta weighted features ? This was not clear to me. I can see that you can partition, the variance explained among your features and that those add up to 100% but I am not clear on what you are predicting at this point.

Similarly, the methods could use more equations so as to be crystal clear. For this review, I read the King et al. Neuroimage paper on B2B and it really helped me understanding what you were doing. I found the equations very helpful and the analytical description of the methods in general more understandable. You don't want to repeat what's in that paper but since Jean-Rémi is a co-author, you could use similar notation, and in any case expand...

Minor Points.

I. 66 in results. Please state that chance level is 50%.

In figure 1. How about using the same scale for B and C. I think it would further help you make your point.

In figure 2. Would it be possible to use the same "normalized" scale for these beta coefficients? How can a Beta coefficient be above 1? (the scale says 150 for acoustic...)

I. 67. Since you are flipping between B2B and encoding, I am not sure at this point that a reader would understand what variance you are referring to and how you estimate your ceiling value. This should be stated more explicitly.

Figure 3. The upper panel is not really a cumulative decoding performance, just the decoding at the diagonal no? As I mentioned in my prior review, a cumulative decoding assessment would be interesting too...

I. 515 in methods – the upper frequency is 11,250 Hz not 1,125 Hz.

I. 548 in methods. "From this, we use the beta-coefficients that map the true stimulus feature to the predicted.." - I am not sure that everyone is going to understand that at this stage you are using only a subset of the regression coefficient (i.e. the "diagonal"). Maybe add one more sentence to explain this. Also maybe it should say " the beta coefficient" in singular instead of plural. You are only using the one entry in that vector of coefficients, correct?

I found the supplemental section very hard to read (honestly I gave up). That might be ok and I realize that it is in part responses to our reviews but it might really benefit from a road map; maybe a short intro that states the various goals in these supplemental analysis right at the beginning and where one can find them in that long series of figures.

Frederic Theunissen

Reviewer #5:

Remarks to the Author:

I have joined the review of this manuscript at this juncture with the specific remit to assess how the authors have responded to the comments of Reviewer 4.

In general, I think they have done a very thorough and satisfactory job in responding to the pointwise comments by that reviewer. That said, I think some of the overarching comments from that reviewer on the significance of the advance over previous research have gone unaddressed.

In addition, I have some comments/questions of my own.

- 1) My main overall comment is with regard to the presentation of the work. I think the manuscript is rich in valuable ideas and clever analyses. And I thought the introduction and discussion were very well written and interesting. However, I found the results section very dense and somewhat dry. My opinion (which I am sure you will prefer to ignore) is that this would have been better as a longer form paper. I think burying the methods at the end of the paper does a disservice to its readability and, ultimately, its impact. I have significant experience using encoding/decoding methods for speech neuroscience and I found the results section fairly torturous. If the word limits are not so strict – my main suggestion would be to breathe some life back into the results section by adding some appropriately placed motivating sentences, additional sentences providing more intuition about each of the methods, and definitions of certain of the more arcane terms.
- 2) Apologies if I missed it – but what is the interpretation for the data in Fig 2, where it appears that the neural decoding of phoneme position gets worse from P1 – P5, but then gets better again for P-5 to P-1?
- 3) I don't understand how the x-axis in Fig 2B goes beyond 4 phonemes when the analysis involved simulating responses to 4-phoneme anagrams.
- 4) Emphasizing that the TG analysis was conducted at a single time lag (train or test) in the main body would be helpful in understanding how to interpret Fig 3. (Again, referring back to point 1 – this is an example of where burying the methods at the end does a disservice to the readability of the main body of the paper- IMO).
- 5) I would move the top panel of Fig 3 below the bottom panel. At the moment, the bottom panel is always discussed first. So why not make it the first subfigure.
- 6) I don't know what "(superimposed)" means in the phrase "superimposed" word onset in the caption of Fig 3.
- 7) I thought the sentence "The representational trajectories (Figure 3) complete a full evolution cycle every ~80 ms (i.e. average phoneme duration)" was unnecessarily abstract.
- 8) Section 1.10.1 perfectly epitomizes my point number 1 above.
- 9) I may be misunderstanding – but I found it difficult to understand how the following two things can be true at the same time: (i) phonemes that are early in a word are less predictable and decodable earlier, while (ii) more predictable phonemes are decoded earlier. I guess there is an interaction going on?

We are extremely thankful to the reviewers for their time and for sharing their thoughtful insight into our work. The manuscript has significantly improved due to incorporating their suggestions into the article. The primary changes since the previous submission include the following:

[1] Reviewers 1 and 5 correctly point out that the previous version of the manuscript was not concrete and explicit enough regarding the novelty and impact of the current work as it pertains to elucidating the computations supporting speech comprehension in general, as well as its advances over previous studies more specifically. Our changes include dialing back certain claims regarding the link between phonetic processing and lexical processing, as well as making clear what results were already known from previous work, what results are new from this work, and what the implications of the novel findings are for speech comprehension. We have also addressed some of the related concerns raised by reviewer 4 in the previous round.

[2] It was also correctly highlighted that there are two key limitations of the current work, namely that the effect sizes of phonetic decoding was quite low, and that we do not have an online measure of speech comprehension to be able to directly relate the decoding of speech features with speech understanding - we instead have comprehension questions every few minutes to keep participants engaged, at which participants performed at ceiling. We make these shortcomings clear to the reader at multiple points in the manuscript, including both the results and discussion sections. We agree that these are important next steps for future work to address, but believe that these limitations do not challenge our main findings, or the main claims we are making in the current paper. Specific changes, and how they relate to specific suggestions from the reviewers are provided below.

[3] The present paper comprises a large number of complex and novel analyses. Reviewers 3 and 5 correctly point out that this caused the previous version of the manuscript to be difficult to follow, particularly in the results, methods and supplementary sections. To address this, we have substantially edited the results section by adding motivating sentences at the beginning of each new analysis result, and explain the approach and findings in a much more palatable manner. We believe that this makes the results section much easier to follow and appreciate. Furthermore, we have added a number of mathematical equations as formal annotation of the analysis steps to the materials section, which will aid the reader both in understanding our approach and implementing it in their own work. Finally, we have added a contents page to the supplementary materials and a series of informative subheadings to help guide the reader through the numerous additional analyses. Again, the specifics of our edits are described in detail below.

[4] Finally, both reviewers 3 and 5 pointed out a number of instances in the manuscript that were confusing or ambiguous. We thank the reviewers for carefully going through the manuscript and bringing these points to our attention. We believe that we have sufficiently addressed all of these confusions.

Below, we address each reviewer's comments point-by-point.

Reviewer #1 (Remarks to the Author):

I would like to thank the authors for their efforts in responding in detail and with great clarifications and new analyses to my more technical concerns, most of which have been resolved.

I value the technical level of detail with which the authors have addressed comments, and I found many if not most of my more technical concerns about potential confounds addressed satisfactorily. (But certainly not all, see esp. the chosen non-use of Bayes Factors for null effects.)

However, this is also to note that the authors' response has remained rather deflective of my and others' more substantial concerns, esp. on this study's very limited ability to elucidate speech and language perception and behaviour per se and, related, the low decoding accuracies. Both have been covered in a brief paragraph in the discussion, and I will leave it to the editors' judgement whether this is deemed sufficient.

Put most succinctly, statements like "[this study] shed new light on how the human brain combines them to probe the mental lexicon" (abstract, concluding remarks) strike me as not really addressed by the study and analyses at hand. I congratulate the authors on a technical tour de force but remain uncertain about the larger implications and relevance of this work.

Thank you again for your very helpful comments on the manuscript. We agree that we may have overstepped in stating implications for our understanding of the mental lexicon more generally, and have walked these claims back where relevant. In addition, we recognize that we have not made our specific claims clear in the manuscript, and have resolved this in the updated version. The main changes include:

- [1] Revising the final sentences of the abstract
- [2] Revising the concluding statement of the paper
- [3] Providing provisions in the results sections where decoding performance and behavior is of relevance
- [4] Expanding our commentary of the shortcomings of the study in the discussion
- [5] Adding a paragraph to the beginning of the discussion explicitly stating what the larger implications of our work are, and how they build upon previous findings

[1] The final sentence of the abstract now reads:

“Our results reveal how phonetic sequences in natural speech are represented at the level of populations of neurons, providing new insight into what intermediary representations exist between sensory input and sub-lexical units. The flexibility in the dynamics of these representations paves the way for further understanding of how such sequences may be used to interface with higher order structure such as lexical identity.”

[2] The final sentence of the discussion now reads:

“Our results reveal that the brain implements an extremely elegant computational solution to the processing of rapid, overlapping phoneme sequences: the phonetic content of the unfolding speech signal is jointly encoded with elapsed processing time, thus representing both content and order without relying on a position-specific coding scheme. The result is a sliding phonetic representation of the most recently heard speech sounds. The temporal dynamics of these computations are strikingly flexible, and vary as a function of certainty about both phonological and lexical identity. Overall, these findings provide a critical piece of the puzzle for how the human brain parses and represents continuous speech input, and links this input to stored lexical identities.”

[3] Multiple provisions have been included in the results:

[a] “For reference, the average significant decoding accuracy is extremely modest, at around 51-52%, where chance level is 50%. The effect sizes being dealt with here, therefore, are very small.”

[b] “This thus distinguishes sequences composed of the same phonemes such as ‘pets’ versus ‘pest’, because the features of the speech sounds are ‘time-stamped’ with the relative order with which they entered into the system. A crucial question to be addressed in future work is whether behavioural confusions in language perception arise from errors in these time-stamps.”

[c] “This avoids representational overlap between the phonetic features of neighbouring speech sounds. Future work would benefit from linking behavioural difficulty in speech comprehension with the extent of representational overlap, to assess the existence of a causal relationship.”

[4] We have expanded the section in the discussion regarding shortcomings of our results:

“Our study has several shortcomings that should be addressed in future work. First, the poor signal-to-noise ratio of single-trial MEG lead to very low decoding performances in our results - peaking only 1-2% greater than chance in most cases, on average. The results we report are detectable because we collected responses to thousands of speech sounds. This limits our ability to make claims about the processing of specific speech features, within specific contexts, which do not occur often enough in our dataset to sufficiently evaluate. Future work would benefit from recording a greater number of repetitions of fewer speech sounds to allow for signal aggregation across identical trials. Second, the spatial resolution and the signal-to-noise ratio of MEG remain insufficient to clearly delineate the spatial underpinning of the dynamic encoding scheme. Understanding how the location and configuration of neural responses to speech sounds evolves as a function of elapsed processing time will require data with both high spatial and temporal resolution, such as electro-corticography recordings from the cortical surface. Finally, in part to address the first limitation, we chose a passive listening paradigm, in order to optimise the number of trials we could obtain within a single recording session. This prohibits us from associating decoding performance with behaviour and task performance, which is a key next step for understanding the link between speech representations and speech understanding. A number of interesting hypotheses can be tested in this regard. For instance, are lapses in comprehension explained by poorer decoding of phonetic input? If the brain fails to transform information fast enough along the processing trajectory, does this lead to predictable errors in perception due to overlapping representations? Causally relating the representational trajectories we observe to successful and unsuccessful comprehension will be critical for further understanding the role of these computations for speech processing.”

[5] We have added the following two paragraphs to the beginning of the discussion, which outlines our main claims and their importance for speech processing:

“How the brain processes sequences is a major neuroscientific question, fundamental to most domains of cognition (Euston, 2007; Dehaene, 2015). Precise sequencing is particularly vital and central in speech comprehension: Sensory inputs are transient and unfold rapidly, yet the meaning they convey must be constructed over long timescales. While much is known about how the features of individual sounds are processed (Mesgarani et al., 2014; Khalighinejad et al., 2017), the neural representation of speech *sequences* remains largely unexplored.

Here we analyse neural responses of human participants listening to natural stories, and uncover three fundamental components of phonetic sequencing. First, we add specificity to the claim that the brain does not process and discard inputs at the same rate with which new inputs are received (Gwilliams et al., 2018). Namely, we find that the phonetic representations of the three most recently heard phonemes are maintained in parallel. Second, we show how the brain reconciles inputs that unfold faster than associated neural processing: The content of a speech sound is jointly encoded with the amount of time elapsed since the speech sound began. This encoding scheme is what allows the brain to represent the features of multiple phonemes at the same time, while housing them within distinct activity patterns. Third, we observe that the timing of initiation and termination of phonetic processing is not fixed. Rather, processing begins earlier for more predictable phonemes in the sequence, and continues longer when lexical identity is uncertain. This has the critical implication that phonetic processing, phoneme sequence processing and lexical processing are engaged in continuous interaction. Our results provide new insight into the temporal dynamics of auditory sequence representations and associated neural computations, and pave the way for understanding how such sequences in speech index higher order information such as lexical identity.”

Reviewer #2 (Remarks to the Author):

In this resubmission of "Neural dynamics of phoneme sequences: Position-invariant code for content and order", Gwilliams and colleagues use MEG data during natural story listening to decode the phonetic content over different sliding windows, and show that multiple phonemes (in order) can be decoded from each time sample. In the resubmission, they were responsive to my and the other reviewer's comments, and the clarity of the manuscript has improved. I appreciated the additional analyses showing the spatial evolution of phoneme processing. Although the decoding accuracy is still relatively low, they do show overall that noninvasive imaging can be used to decode phoneme sequences in natural speech. Overall, I am satisfied with their responses to the reviews.

Thank you very much for your time and constructive comments.

Reviewer #3 (Remarks to the Author):

The authors have diligently addressed most of the comments and criticism from the previous review cycle. Their description of this instantaneous yet dynamical code for triphones is novel and interesting. The analyses are rigorous and complete. In the revised version, the authors have added some important controls that further validate the relevance of this dynamic representation of phonemes for linguistic processing. My major complaint: the methods are somewhat complicated, and the results section is written in a very concise form. As a result the paper is not an easy read. I think that the authors could further describe their methodology in the methods section; for e.g. by including all equations with clearly labeled variables. See also my comment below.

Thank you for this suggestion. We have expanded the methods section, and also made the results section more palatable based on a similar suggestion from Reviewer 5.

Major Point.

I applaud the use of effect size instead of t-values (or raw Beta coefficient values) but the calculation of the “variance explained” effect size is missing from the paper: I could not find it in the results or in the methods (or in the response to the reviewers). I suspect that this is something that might be obvious to the authors but not so much so for readers who are trying to follow the sequence of regression analyses in B2B decoding: Using the 31 features, what variance are you trying to capture? The variability in the decoded features using the actual features? The variability in the MEG signal using beta weighted features? This was not clear to me. I can see that you can partition the variance explained among your features and that those add up to 100% but I am not clear on what you are predicting at this point.

Thank you for pointing this out. We have clarified both in the results and methods section how we compute variance explained.

In results:

“The effect sizes we presently report correspond to the proportion of variance explained by each feature according to the B2B model, and relative noise ceiling estimate across all features. The noise ceiling is estimated as the maximum amount of variance the model was able to explain at any latency. We compute this by summing beta coefficients across all features and taking the maximum over time. Each feature timecourse is then normalised by this maximum performance measure, to provide a proportion of variance explained”.

In methods:

“To aid interpretability of our effect sizes, we computed the proportion of variance explained for each feature. To obtain this measure, we analyze the beta coefficients that map each true feature f to the target feature at each time sample t . Specifically, we first compute a noise ceiling, by summing the beta coefficients across all features, and taking the maximum value at any latency:

$$\hat{R}_{ceiling}^f = \max_t \sum_{f=1}^{f=31} \beta_f \quad (5)$$

$\hat{R}_{ceiling}$ thus represents the upper limit of variance our model can explain, taking all features together. We then normalise the beta coefficient time-courses of each feature by this upper limit value:

$$\hat{R} = \hat{R}_{B2B} / \hat{R}_{ceiling} \quad (6)$$

The result is a proportion of variance explained, relative to the contribution of all features in the model.”

Similarly, the methods could use more equations so as to be crystal clear. For this review, I read the King et al. Neuroimage paper on B2B and it really helped me understanding what you were doing. I found the equations very helpful and the analytical description of the methods in general more understandable. You don't want to repeat what's in that paper but since Jean-Rémi is a co-author, you could use similar notation, and in any case expand...

We have now clarified the B2B method using equations and similar notation to Jean-Remi's Neuroimage paper:

“In short, B2B consists of both a decoding and an encoding step: (1) the decoding model G_f aims to find the combination of channels that maximally decode feature f and (2) the encoding model H_f estimates whether the decoded predictions \hat{y}_f are specific to f and/or attributable to other, covarying, features.

To implement B2B, we first fit a ridge regression model on a random (shuffled) 50% split of the epoched data. A decoding model $G_f \in \mathcal{R}^{C \times F}$ was trained across the $C=208$ MEG channels for each of the $F=31$ phonetic features independently at each of 201 time-points in the epoch. That is, we train and test a unique decoder at each time sample. The mapping was learnt between the multivariate input (activity across sensors) and the univariate stimulus feature (one of the 31 features described above). All decoders were fit on data normalised by the mean and standard deviation in the training set:

$$\operatorname{argmin}_G \sum_i (y_i - G^T X_i)^2 + \alpha_G \|G\|^2 \quad (3)$$

where $y_i \in \{\pm 1\}$ is the feature to be decoded at trial i and X_i is the corresponding MEG activity. The l2-regularisation parameter α_G was also fit, testing 20 log-spaced values from 1^{-5} to 1^5 . This was implemented using the `RidgeCV` function in `scikit-learn` citep{pedregosa2011scikit}.

Then, we use the remaining 50% split of data to train the encoder $H \in \mathcal{R}^{F \times F}$ for each of the decoded features. To do this, we fit a second ridge regression model to estimate the coefficients that map the `true` features onto the `decoded` features:

$$\operatorname{argmin}_H \sum_i (Y_i - H^T G^T X_i)^2 + \alpha_H \|H\|^2 \quad (4)$$

where Y_i in $\{\pm 1\}$ are the true features of a given phoneme at trial i and $G^T X_i = \hat{Y}_i$ is the features decoded from the MEG at that trial. Again, a regularisation parameter α_H was learnt for this stage with `textit{RidgeCV}`.

This second step of B2B allows the models to estimate whether the decoded features f is solely explained by its covariates. In particular, King et al. (2020) showed that, under Gaussian and heteroscedasticity, the $H_{\{f\}}$ tends a positive value if and only if f is linearly and specifically encoded in the brain activity and not reducible to its covarying features.

From this, we use the "beta" coefficient that maps the true stimulus feature to the predicted stimulus feature ($\text{diag}(H)$) as a metric of `specific` decoding performance. If a stimulus feature is not encoded in neural responses (the null hypothesis) then there will be no reliable mapping between the true feature y and the model prediction \hat{y} . Thus, the beta coefficient will be indistinguishable from zero -- equivalent to chance performance. If, however, a feature `is` encoded in neural activity (the alternative hypothesis), we should uncover a significant relationship between y and \hat{y} , thus yielding an above-zero beta coefficient."

Minor Points.

I. 66 in results. Please state that chance level is 50%.

Thank you, we have made this change.

In figure 1. How about using the same scale for B and C. I think it would further help you make your point.

Thank you, we have made the scales more similar for the information theoretic features and the position features.

In figure 2. Would it be possible to use the same "normalized" scale for these beta coefficients? How can a Beta coefficient be above 1? (the scale says 150 for acoustic...)

Good idea. We now plot normalized decoding across the neural and acoustic analysis, and across phoneme positions.

I. 67. Since you are flipping between B2B and encoding, I am not sure at this point that a reader would understand what variance you are referring to and how you estimate your ceiling value. This should be stated more explicitly.

We now include a much more detailed description of how we compute the ceiling, provided above. When referring to variance explained from B2B, we now use $R_{\hat{}}$ notation to distinguish it from encoding R .

Figure 3. The upper panel is not really a cumulative decoding performance, just the decoding at the diagonal no? As I mentioned in my prior review, a cumulative decoding assessment would be interesting too...

Right, it is cumulative over phoneme positions but not within a phoneme position. We have clarified the panel caption: "Decoding performance of just the diagonal axis of each phoneme position (where train times are equal to test times). The visualisation therefore represents when phonetic information is available, regardless of the topography which encodes it. We use a stack plot, such that the variance explained by different phoneme positions is summed along the y-axis."

I. 515 in methods – the upper frequency is 11,250 Hz not 1,125 Hz.

Thank you for catching this mistake. We have fixed it.

I. 548 in methods. "From this, we use the beta-coefficients that map the true stimulus feature to the predicted.." – I am not sure that everyone is going to understand that at this stage you are using only a subset of the regression coefficient (i.e. the "diagonal"). Maybe add one more sentence to explain this. Also maybe it should say " the beta coefficient" in singular instead of plural. You are only using the one entry in that vector of coefficients, correct?

Thank you for pointing out this potential source of confusion. We actually perform this procedure for all train/test pairs (the full TG matrix) and then use the beta coefficients both of the diagonal and of the full matrix in our analyses. We have clarified this point in the methods:

“This procedure was applied at all possible train/test time combinations for each time sample from -200 to 600 ms relative to phoneme onset. For the results shown in Figures 1, 2 and 4, we just analyse the beta coefficients where the train time is equal to test time (the `diagonal’). Whereas the analyses shown in Figures 3 and 5 use the full 2-dimensional timecourse of decoding performance.”

I found the supplemental section very hard to read (honestly I gave up). That might be ok and I realize that it is in part responses to our reviews but it might really benefit from a road map; maybe a short intro that states the various goals in these supplemental analysis right at the beginning and where one can find them in that long series of figures.

Thank you for the suggestion. We have now added an introductory paragraph which states the four main types of additional analyses we conducted. We then organize each analysis under one of four sub-headings, along with a table of contents to the supplementary material. For each section of the supplementary methods, we provide another short paragraph which motivates the analysis and summarizes the result. In the final publication, we will ask to add hyperlinks between the relevant sections of the roadmap and the figures in question.

Frederic Theunissen

Reviewer #5 (Remarks to the Author):

I have joined the review of this manuscript at this juncture with the specific remit to assess how the authors have responded to the comments of Reviewer 4.

In general, I think they have done a very thorough and satisfactory job in responding to the pointwise comments by that reviewer. That said, I think some of the overarching comments from that reviewer on the significance of the advance over previous research have gone unaddressed.

Thank you for highlighting this point. This was also raised by Reviewer 1. We have now added two paragraphs in the discussion which makes clear the state of previous knowledge, what our concrete findings are, and how we believe they advance the field:

“How the brain processes sequences is a major neuroscientific question, fundamental to most domains of cognition (Euston, 2007; Dehaene, 2015). Precise sequencing is particularly vital and central in speech comprehension: Sensory inputs are transient and unfold rapidly, yet the meaning they convey must be constructed over long timescales. While much is known about how the features of individual sounds are processed (Mesgarani et al., 2014; Khalighinejad et al., 2017), the neural representation of speech *sequences* remains largely unexplored.

Here we analyse neural responses of human participants listening to natural stories, and uncover three fundamental components of phonetic sequencing. First, we add specificity to the claim that the brain does not process and discard inputs at the same rate with which new inputs are received (Gwilliams et al., 2018). Namely, we find that the phonetic representations of the three most recently heard phonemes are maintained in parallel. Second, we show how the brain reconciles inputs that unfold faster than associated neural processing: The content of a speech sound is jointly encoded with the amount of time elapsed since the speech sound began. This encoding scheme is what allows the brain to represent the features of multiple phonemes at the same time, while housing them within distinct activity patterns. Third, we observe that the timing of initiation and termination of phonetic processing is not fixed. Rather, processing begins earlier for more predictable phonemes in the sequence, and continues longer when lexical identity is uncertain. This has the critical implication that phonetic processing, phoneme sequence processing and lexical processing are engaged in continuous interaction. Our results provide new insight into the temporal dynamics of auditory sequence representations and associated neural computations, and pave the way for understanding how such sequences in speech index higher order information such as lexical identity.”

Here we also paste previous comments of reviewer 4 and address them point-by-point:

While the findings on phoneme surprisal and lexical entropy are potentially interesting (although the context can be improved, for example the relation with another MEG study by Brodbeck & Simon, 2018), not much new can be concluded from the preceding analyses.

Previous studies, including Brodbeck and Simon (2018) have shown that the strength of phonetic encoding is modulated by surprisal and entropy. To date, no study has before shown that surprisal and entropy modulate the *latency* of phonetic encoding. Critically we show that phonetic processes are initiated earlier when the sound was predicted, and terminated later when the word identity is unknown. This is a novel finding that has never been reported before, and has crucial implications for the bi-directional interaction between phonetic and lexical processing.

The relevant part of the discussion explains this result: “Although previous work has shown that processing of the speech signal is sensitive to phoneme probability within a word (Gagnepain et al., 2012; Gwilliams and Marantz, 2015; Gwilliams et al., 2017; Brodbeck and Simon, 2018; Di Liberto et al., 2019; see Gwilliams and Davis, 2022 for a review) this is the first study quantifying the *consequences* this has for encoding the phonetic content of those phonemes. Interestingly, we did not observe an effect of predictability on overall decoding performance, suggesting that processing delays may serve as a compensatory mechanism to allow more information to be accumulated in order to reach the overall same strength of encoding (Gold and Shadlen, 2007).”

The fact that phoneme decoding does not generalize across time has been shown before and likely reflects the fact that responses propagate from region to region in the auditory system – a basic fact that is well established.

Our analyses on the sensor data show that, in direct contrast to the hierarchical assumption that information is passed from low to high level areas, information remains locally processed within auditory regions. It appears to change its configuration locally over time, rather than moving anteriorly with time, for instance. This is a novel and important observation, and is suggestive of local recurrent computations rather than feedforward hierarchical computations. The implication is that the entire sequence remains encoded within auditory areas, rather than having different elements of the sequence be globally spatially distributed.

The relevant part of the discussion explains this result: “The processing trajectory we find for human speech processing did not trace a spatial path across distinct regions. Instead, phonetic features remained encoded within auditory regions for around 100–400 ms, and the trajectory was different for different features. This suggests that information may be locally changing its population-level configuration within the auditory cortices rather than following a strict anatomical transposition from low to high level areas (Yi et al., 2019; Millet et al., 2021; King et al., 2021)”.

The fact that this generalization falls below the $p < 0.05$ level at about the average duration of a phoneme also seems hard to interpret. This cutoff is arbitrary and will be affected by the SNR of the data.

We have conducted a number of additional analyses whereby we modify the SNR of the neural data, either by adding noise or reducing trial counts, and show that the processing dynamics, and resulting conclusions, remain unaltered. The many analyses to this point are included in the supplementary materials.

And of course, MEG is a very coarse measure of the neural response that pools across multiple different brain regions, each of which probably have a different time constant. What is most relevant is the time constant of the single neuron responses within the regions of auditory cortex that the brain is decoding from – who knows how this relates to what is being measured here.

Our results inform how populations of neurons, likely with different time constants, serve to encode phonetic and temporal information. We have made the spatial scale of our claims clear in the abstract: “Our results reveal how phonetic sequences in natural speech are represented at the level of populations of neurons, providing new insight into what intermediary representations exist between the sensory input and sub-lexical units.”

The results of applying the cross-decoding to the spectrogram (Fig 1D) do not make much sense. Phonemes clearly have acoustic cues that vary with their onset and offset even if their acoustic realization is modulated substantially by contextual factors. Figure 1D seems to imply that one would not be able to distinguish the phonemic content over ~300 millisecond window, which is clearly not true.

That the decoding on the acoustic signal of the 20,000 phoneme instances we entered into the model is stable for 300 ms shows that there exist stable and reliable cues for phonetic features in the auditory signal, even though the signal itself evolves over time.

This result makes question the interpretation of the neural data. The fact that decoding is more sustained for longer phonemes (Fig 3A) and is at least partially consistent across position (Fig 3B) is not surprising as there are features that are shared across these contexts.

Position-specific encoding has been put forward in previous work as a hypothesis for how speech sounds are represented. Here we show that, indeed, that is not the case. We provide an extensive discussion on this point on lines 382-394.

In addition, I have some comments/questions of my own.

1) My main overall comment is with regard to the presentation of the work. I think the manuscript is rich in valuable ideas and clever analyses. And I thought the introduction and discussion were very well written and interesting. However, I found the results section very dense and somewhat dry. My opinion (which I am sure you will prefer to ignore) is that this would have been better as a longer form paper. I think burying the methods at the end of the paper does a disservice to its readability and, ultimately, its impact. I have significant experience using encoding/decoding methods for speech neuroscience and I found the results section fairly torturous. If the word limits are not so strict – my main suggestion would be to breathe some life back into the results section by adding some appropriately placed motivating sentences, additional sentences providing more intuition about each of the methods, and definitions of certain of the more arcane terms.

Thank you very much for this suggestion. We have revised the results section significantly to make it more palatable to the reader:

Example 1: “Having demonstrated that the brain processes multiple speech sounds at the same time, the next question is: How does the brain do this without mixing up the phonetic features of these speech sounds? There are a number of potential computational solutions to this problem. One is position-specific encoding, which posits that phonetic features are represented differently depending on where the phoneme occurs in a word. This coding scheme uses a different neural pattern to encode information about the first phoneme position (P1), second phoneme position (P2), etc., resulting in no representational overlap between neighbouring speech sounds.

To test whether the brain uses this coding scheme”

Example 2: “The sustained, parallel and position-invariant representation of successive phonemes leaves our original question unanswered: How does the brain prevent interference between neighbouring speech sounds?

A second possible solution to this problem is that, rather than embedding phonetic features within a different neural pattern depending on phoneme location, phonetic features are embedded in different neural patterns as a function of latency since phoneme onset. This means that each speech sound travels along a processing trajectory, whereby the neural population encoding that speech sound evolves as a function of elapsed processing time.

To test whether this was the case”

Example 3: “The neural pattern that encodes phonetic features evolves with elapsed time since phoneme onset. What purpose does this evolution have? We sought to test whether the evolution is systematic enough to explicitly encode latency since the speech sound began. If latency information is indeed encoded along with phonetic detail, the consequence would be an implicit representation of phoneme order in addition to phoneme identity. This is because the brain would have access to that fact that features of phoneme X occurred longer ago than than phoneme X+1, and those, longer ago than phoneme X+2, etc.

To formally test this possibility...”

Example 4: “Next we asked whether another purpose of this dynamic coding scheme is to minimise overlap between neighbouring speech-sound representations. To test this, we quantified the extent to which successive phonemes are simultaneously represented in the same neural assemblies.”

Example 5: “Is this dynamic coding scheme also present in the auditory signal, and so all of this comes for free by virtue of speech being a dynamic input? Or, is dynamic encoding the result of a specific transformation the brain applies to that input? To address this issue...”

2) Apologies if I missed it – but what is the interpretation for the data in Fig 2, where it appears that the neural decoding of phoneme position gets worse from P1 – P5, but then gets better again for P-5 to P-1?

The overall decoding strength decreases with distance from word onset/offset because fewer trials are being entered into the analysis. In response to a suggestion from Reviewer 3, we now plot these decoding time-courses normalized across phoneme position.

3) I don't understand how the x-axis in Fig 2B goes beyond 4 phonemes when the analysis involved simulating responses to 4-phoneme anagrams.

We simulate the responses to anagrams as if they existed within continuous speech. So phoneme positions 5 and 6 exist across word boundaries. We have now made this clearer in the text: "...we simulated the neural responses to continuous sequences of 4-phoneme anagrams using the decoding model coefficients to reconstruct synthesised MEG responses...". And in the relevant figure caption: "Result of simulating and reconstructing phoneme sequences from the MEG model coefficients. x-axis represents how many previous phonemes can be reconstructed; y-axis represents the cosine similarity between the true phoneme sequence and the reconstruction. History lengths exceeding 4 phonemes correspond to the phonemes of the previous word in the sequence."

4) Emphasizing that the TG analysis was conducted at a single time lag (train or test) in the main body would be helpful in understanding how to interpret Fig 3. (Again, referring back to point 1 – this is an example of where burying the methods at the end does a disservice to the readability of the main body of the paper- IMO).

Thank you. We have now clarified this point:

"This method consists of learning the most informative spatial pattern for a phonetic feature at a given moment, and assessing the extent to which it remains stable over time. TG results in a training X testing matrix that helps reveal how a neural representation evolves over time. This analysis is applied relative to phoneme onset."

5) I would move the top panel of Fig 3 below the bottom panel. At the moment, the bottom panel is always discussed first. So why not make it the first subfigure.

This is a great suggestion. We have made this change, thank you.

6) I don't know what "(superimposed)" means in the phrase "superimposed" word onset in the caption of Fig 3.

We agree that this word is cumbersome and does not add meaning, and thus agree to remove it. Thank you.

7) I thought the sentence "The representational trajectories (Figure 3) complete a full evolution cycle every ~80 ms (i.e. average phoneme duration)" was unnecessarily abstract.

We agree and have changed that paragraph to:

"Although phonetic features are present in neural responses for around 300 ms (length of the decoding 'finger', in Figure 3) any given spatial pattern of activity which encodes that information (width of the decoding 'finger') is informative for much shorter than that – about 80 ms. We noted that 80 ms is very similar to the average duration of the phonemes in our stories (M=78 ms, SD=34 ms). A crucial question is whether the speed of evolution is fixed, or whether it scales with phoneme duration and speech rate."

8) Section 1.10.1 perfectly epitomizes my point number 1 above.

We have simplified this section and sections like it, to make it less dense.

9) I may be misunderstanding – but I found it difficult to understand how the following two things can be true at the same time: (i) phonemes that are early in a word are less predictable and decodable earlier, while (ii) more predictable phonemes are decoded earlier. I guess there is an interaction going on?

Thank you for pointing this out. This was a typo on our part. Phonemes that are early in the word are less predictable, and are decoded later, not earlier.

Reviewers' Comments:

Reviewer #3:

Remarks to the Author:

You have very thoroughly addressed all of my comments. This is a novel and important contribution to our understanding of the neural representation of phoneme sequences.

Reviewer #5:

Remarks to the Author:

I think the authors have done a commendable job in responding to my - and the other reviewers' - comments. I think this is a paper with many interesting ideas and analyses and that it will be of interest to many researchers interested in speech processing in the human brain.